# Spatial variation and geographical weighted regression analysis to explore open defecation practice and its determinants among households in Ethiopia

**Nebiyu Mekonnen Derseh** *, **Meron Asmamaw Alemayehu, Muluken Chanie Agimas, Getaneh Awoke Yismaw, Tigabu Kidie Tesfie, Habtamu Wagnew Abuhay**

Department of Epidemiology and Biostatistics, Institute of Public Health, College of Medicine and Health Sciences, University of Gondar, Gondar, Ethiopia

* nebiyumek12@gmail.com

**Data Availability Statement:** All of the included data is available in the paper. For further inquiries, DHS data cannot be shared publicly because of

## Abstract

### Background

In Ethiopia, recent evidence revealed that over a quarter (27%) of households (HHs) defecated openly in bush or fields, which play a central role as the source of many water-borne infectious diseases, including cholera. Ethiopia is not on the best track to achieve the SDG of being open-defecation-free by 2030. Therefore, this study aimed to explore the spatial variation and geographical inequalities of open defecation (OD) among HHs in Ethiopia.

### Methods

This was a country-wide community-based cross-sectional study among a weighted sample of 8663 HHs in Ethiopia. The global spatial autocorrelation was explored using the global Moran's-*I*, and the local spatial autocorrelation was presented by Anselin Local Moran's-I to evaluate the spatial patterns of OD practice in Ethiopia. Hot spot and cold spot areas of OD were detected using ArcGIS 10.8. The most likely high and low rates of clusters with OD were explored using SaTScan 10.1. Geographical weighted regression analysis (GWR) was fitted to explore the geographically varying coefficients of factors associated with OD.

### Results

The prevalence of OD in Ethiopia was 27.10% (95% CI: 22.85–31.79). It was clustered across enumeration areas (Global Moran's $I$ = 0.45, Z-score = 9.88, P-value $\leq$ 0.001). Anselin Local Moran's I analysis showed that there was high-high clustering of OD at Tigray, Afar, Northern Amhara, Somali, and Gambela regions, while low-low clustering of OD was observed at Addis Ababa, Dire-Dawa, Harari, SNNPR, and Southwest Oromia. Hotspot areas of OD were detected in the Tigray, Afar, eastern Amhara, Gambela, and Somali regions. Tigray, Afar, northern Amhara, eastern Oromia, and Somali regions were explored as having high rates of OD. The GWR model explained 75.20% of the geographical variation of OD among HHs in Ethiopia. It revealed that as the coefficients of being rural residents,

legal restrictions of DHS program. However, all of the included data can be found on the following websites: https://dhsprogram.com/data/available-datasets.cfm.

**Funding:** The author(s) received no specific funding for this work.

**Competing interests:** No authors have competing interests.

**Abbreviations:** AICc, corrected Acacias information criterion; ArcGIS, Geographic Information System; CI, confidence interval; CSA, the Central Statistics Agency; DHS, Demographic Health Survey; EAs, Enumeration Areas; EDHS, Ethiopian Demographic and Health Survey; EMDHS, Ethiopian Mini-Demographic and Health Survey; EPHC, Ethiopian population and housing Census; GWR, Geographical weighted regression; HH, Household; LLR, Log likelihood Ratio; LMICs, Low-and Middle-income countries; OD, Open defecation; OLS, Ordinary least square; RR, relative risk; SDG, Sustainable Development Goals; SNNPRP, South Nations Nationalities and People's Region; SSA, Sub-Saharan Africa; VIF, Variance inflation factor; WHO, World Health Organization.

female HH heads, having no educational attainment, having no radio, and being the poorest HHs increased, the prevalence of OD also increased.

## Conclusion

The prevalence of OD in Ethiopia was higher than the pooled prevalence in sub-Saharan Africa. Tigray, Afar, northern Amhara, eastern Oromia, and Somali regions had high rates of OD. Rural residents, being female HH heads, HHs with no educational attainment, HHs with no radio, and the poorest HHs were spatially varying determinants that affected OD. Therefore, the government of Ethiopia and stakeholders need to design interventions in hot spots and high-risk clusters. The program managers should plan interventions and strategies like encouraging health extension programs, which aid in facilitating basic sanitation facilities in rural areas and the poorest HHs, including female HHs, as well as community mobilization with awareness creation, especially for those who are uneducated and who do not have radios.

## Introduction

Open defecation (OD) is defined as when people defecate in open places such as fields, forests, bushes, open bodies of water, beaches, and other open spaces, or with solid waste [1]. Open defecation practice is the main public health concern in low- and middle-income countries (LMICs). The World Health Organization (WHO) suggested that more than 1.5 billion people still do not have private toilets globally; out of these, 419 million still practice OD [2, 3]. Nearly 16 million people practiced OD in Latin America and the Caribbean [4], which was mainly contributed by Haiti, with a prevalence of 20% [5]. In the East Asia and Pacific region, around 40 million, or 2%, of HHs practiced OD [6].

According to World Bank statistics, OD is mainly concentrated in Sub-Saharan Africa (SSA), followed by South Asian regions globally. Recent evidence shows that the burden of OD in the South Asian region is 44%, which is mainly accounted for by India. In India, a recent study revealed that 20% of HHs practiced OD [7, 8]. In south Asian countries, the recent prevalence of OD practice ranged from 1.2% in Bangladesh [9] to 19% in India [10].

In 2022, of the world's 420 million HHs with OD practice, SSA accounted for the largest number among all of the regions in the world, with an estimation of 47%. Open defecation in SSA is usually associated with rural areas and poverty-related problems [11]. Western (31.10%) and Southern (25.55%) African regions were highly affected by OD [12]. In Niger, 72% of households practiced OD in 2021. In Ghana, 44.2% of households were defecating openly in 2016 [13]. In Ethiopia, the prevalence of OD seems to be decreasing (32% in 2016 [14] and 27% in 2019 [15]), but it was higher than the pooled prevalence of the SSA estimate (22.55%) [12].

Open defecation usually leads to immediate and long-term adverse effects on human health. Among its immediate effects, OD is the main source of many faeco-oral transmitted diseases such as cholera, dysentery, typhoid, amoebiasis, giardiasis, schistosomiasis, soil-transmitted helminthiasis, and polio [2, 11, 16]. These infectious diseases become an endemic and vicious cycle in high-burden areas with OD, resulting in a large number of morbidities and deaths, especially among children in LMICs, and also contributing to the spread of antimicrobial resistance [2]. The WHO reported that over 38% of diarrhea-related deaths were directly attributable to poor sanitation in LMICs, accounting for 564,000 deaths and almost 30 million

disability-adjusted life years (DALYs) in 2019 [17]. Open defecation was also associated with adverse pregnancy outcomes like preterm birth and low birth weight [18]. Poor sanitation, like the practice of OD, also results in long-term health and socio-economic impacts. Children living with a high burden of OD are usually stunted [16, 19] due to being repeatedly infested by intestinal worms, having a poor appetite, and losing body fluids during illness, which makes them incompetent for education. Poor sanitation affects human well-being and social and economic development because of long-term impacts like anxiety, the risk of sexual assault, and loss of education and work [2, 20].

Evidence suggests the reason why people practice OD is either due to poverty, which makes it a challenge to build latrines, or a lack of government commitments to help and enforce them. On the other hand, even though toilets are available, people often prefer open defecation [21]. As reported by many scholars, being poor, not attending formal education, having a rural residence, having a female household head, having over five family members, having under-five children, having limited access to drinking water, and having an absence of media exposure were significant determinants of open defecation [12, 21, 22].

The Ethiopian government adopted the health extension program (HEP) both in urban and rural areas to safeguard communities against health-related problems. Community HEP achieved a remarkable change in basic hygiene and sanitation practices in Ethiopia. As a result, the magnitude of OD in Ethiopia decreased from 81.9% [23] in 2000 to 27.1% in 2019 [15]. However, evidence still showed that people prefer OD practice rather than using toilets. In Ethiopia, a recent study showed that over a quarter (27.8%) of households practiced OD despite the presence of latrine facilities [21]. On the other hand, around 1 out of 6 households practiced OD after they had certified open defecation-free status in Ethiopia [24]. The most recent evidence also showed that more than one in four households defecated openly in bush or fields; of this, 35% were in rural areas and 10% in urban areas [15]. Because of this, Ethiopia is not on the right track to achieve SDG-6 global targets in 2030.

Currently, evidence shows that geospatial study is very useful to identify determinants as well as help design strategies and plans with efficient use of resources. There are indicators of regional and urban rural variation of OD practices in Ethiopia. In order to reduce the burden of OD and design interventions locally, the exploration of geospatial variations, identification of high-risk clusters, and hotspot areas, and prediction OD practice even in unobserved areas are very crucial in order to design different interventions locally. Moreover, identifying and mapping geographically varying coefficients of determinants is helpful in designing targeted interventions or solutions against those predisposing factors locally. Geographical weighted regression (GWR) analysis is used to explore geographically varying factors associated with the practice of OD that help design interventions locally.

This study would be an important key to designing interventions and solving determinants of OD regionally and locally at the community level. However, the burden of OD practice, whether it was randomly distributed or clustered across EAs and hotspot areas, was unknown in Ethiopia recently. Moreover, exploring geographically varying coefficients using GWR analysis was not studied in Ethiopia. Therefore, this study aimed to explore the spatial variation and geographical inequalities of open defecation among households in Ethiopia using national survey data.

## Materials and methods

### The study settings

This was a country-wide survey study that was conducted regionally in urban and rural areas of Ethiopia from March 21, 2019, to June 28, 2019, as part of the Ethiopian Mini-Demographic

and Health Surveys (EMDHS) 2019, which was the second EMDHS implemented in Ethiopia [15].

Ethiopia is a landlocked country in the Horn of Africa and lies between the latitudes of 3˚ and 15˚ North and the longitudes of 33˚ and 48˚ East. It has a total area of 1,100,000 km$^2$. There are 11 ethnically and politically autonomous regional states and two administrative cities in Ethiopia. The regions are divided into 68 zones, which are further subdivided into 817 districts, which are then subdivided into about 16,253 kebeles (the lowest locally administered units) [14].

Ethiopia has diverse geographical features, which range from the highest peak at Ras Dejen, which is 4,550 meters above sea level, to the lowest at the Depression in the Afar region, 110 meters below sea level. There is also climatic variation with the topographic features, and the temperature is as high as 47 ˚C in the Afar depression and as low as 10 ˚C in the highlands of Ethiopia. Ethiopia's basic economic source is agriculture, which is the backbone of the national economy [25]. Ethiopia is the second-largest populous country in Africa, with an estimated 120 million people in 2022.

## Study design and period

A countrywide community-based cross-sectional study design was conducted with nationally representative weighted samples of 8,663 households in Ethiopia from March to June 2019.

## Population and eligibility criteria

The source population for this study was all households that were living in the urban and rural areas of Ethiopia at the time of the survey, while the study population was all households that were living in the urban and rural areas of the selected EAs and included in the analysis.

## Data source, sample size, and sampling techniques

The source of data for this study was the DHS program database, which we accessed through the https://www.dhsprogram.com/Data/ website after submitting the project title and justification of the study. A total weighted sample of 8663 households was used for this study. Sampling weight was performed to maintain representativeness because of the non-proportional allocation of the sample size in the different regions with their urban and rural variations, as well as the possible differences in response rates [15].

In EMDHS 2019, each region was stratified into urban and rural areas, which were grouped into 21 sampling strata, and then the sample was selected in two stages. In the first stage, stratified samples of census enumeration areas (EAs) in urban and rural areas were selected with complete household (HH) listings using systematic probability sampling based on the sampling frame of all census EAs created for the 2019 Ethiopian Population and Housing Census (EPHC) that was conducted by the Central Statistical Agency (CSA). In the second stage, households (HH) were selected using the same probability systematic sampling as the selected EAs and were interviewed with a household questionnaire [15].

## Study variables

**Dependent variable.** The outcome variable in this study was open defecation practice, based on WHO/UNICEF JMP [3], which was measured by yes for open defecation (no facility or defecating in bush or field) and no for those who have basic sanitation facilities.

**Independent variables.** Explanatory variables for spatial regression were extracted based on a review of previous literature that was associated with open defecation [15, 19].

**Table 1. A list of sociodemographic and economic determinants of open defecation based on a literature review.**

| Sr. No | Sociodemographic and economic determinants | Categories |
|---|---|---|
| 1. | Age of house head in years | (1) 15–24 (2) 25–34 (3) 35–44 (4) 45–54 (5) ≥ 55 |
| 2. | Sex of household head | (1) Male (2) Female |
| 3. | Household educational attainment | (1) No education (2) Primary (3) Secondary (4) Higher education and above |
| 4. | Has radio | (0) No (1) Yes |
| 5. | Has TV | (0) No (1) Yes |
| 6. | Wealth index | (1) Poorest (2) Poorer (3) Middle (4) Richer (5) Richest |
| 7. | Time to get water with round trip | (1) Water on the premises (2) 30 minutes or less (3) More than 30 minutes |
| 8. | Source of drinking water | (0) Improved (1) Unimproved |
| 9. | Residence | (1) Urban (2) Residence |

Sociodemographic and economic characteristics in the household dataset were considered to be determinants in the spatial regression. The details of these factors are listed below (Table 1). The wealth index is defined as a composite measure of a household's cumulative living standard that was divided into 5 quintiles, which were derived by using principal component analysis [15].

## Data collection and tools

The Ethiopian Mini-Demographic Health Survey data was collected through face-to-face interviews using household-level questionnaires. During the data collection period, households were asked to respond to important socio-demographic and economic characteristics that were associated with open defecation in Ethiopia.

## Data management and analysis

Data extraction, cleaning, recoding, and labeling were done using STATA 17 and Microsoft Excel. Before conducting the analysis, sampling weights for each variable were calculated to account for the strata's unequal probability of selection. Missing variables were managed according to DHS guidelines.

**Spatial analysis.** The Spatial Autocorrelation (Global Moran's *I*) tool was performed using both GPS coordinates and proportions of open defecation (OD) to evaluate whether the pattern of OD explored was clustered, dispersed, or random using ArcGIS Version 10.8. The computed tool showed us Moran's I index value, a z-score, and a p-value to evaluate the significance of that index. Statistically significant positive Moran's I value indicates a geographical clustering of OD, while significant negative Moran's index shows dispersion, and if the value is zero, it shows the random distribution. On the other hand, cluster and outlier (Anselin Local Moran's analysis) identified statistically significant clusters of high values (high-high) or low values (low-low) and outliers in which a high value is surrounded primarily by low values (high-low) and a low value is surrounded primarily by high values (low-high). A statistically significant high positive z-score for a feature indicates that the surrounding features have either high or low values, while a statistically significant low negative z-score for a feature indicates outliers.

Hot spot (Getis-Ord Gi*) analysis was used to identify statistically significant hotspots and cold spots of OD. In a statistically significant hot spot, an area with a high value of OD is surrounded by another area with high values of OD, which shows local spatial clustering of high

values of OD, whereas, in a statistically significant cold spot, an area with a low value of OD is surrounded by another area with a low value of OD, which indicates local spatial clustering of low values of OD.

The ordinary Kriging type of interpolation was performed to predict a high prevalence of OD in unsampled EAs using a linear combination of observations at nearby sampled locations in Ethiopia.

SaTScan cluster analysis was conducted using the Bernoulli model to detect the most likely high- and low-rates of OD using SaTScan 10.1.2. A cluster is considered statistically significant when its log likelihood ratio (LLR) is greater than the standard Monte Carlo critical value with a P value less than 0.05. The SaTScan window with the high LLR test was considered the most likely cluster relative to the global distribution of maximum values. The primary and successive most likely clusters were explored based on the LLR test in the ArcMap.

**Spatial regression.  Ordinary Least Squares (OLS) analysis** is a global linear regression model that is used to estimate coefficients by using a dependent variable and a set of quantitative explanatory variables using ArcGIS software. The OLS explored a positive or negative linear fixed relationship and no relationship between an outcome and one or more explanatory numerical variables. The OLS is a single linear model, which can be described in one equation for all features: $Y = \beta_0 + \beta_1 X_1 + \beta_2 X_{2 + \ldots} \beta_n X_n + \varepsilon$; where: Y = dependent variable; $\beta_0$ = intercept; $\beta_1, \beta_2 \ldots \beta_n$ = coefficients; X1, X2...$X_n$ = explanatory variables; $\varepsilon$ = residual. The OLS regression analysis depends on the following six assumptions to be a better fit: Each explanatory variable should have the relationship as we expect; each explanatory variable should be statistically significant either positively or negatively; residuals should not be clustered in location, and this was tested using the global spatial autocorrelation tool; the residuals should be normally distributed and tested using the Jarque-Bera test; each variable should have a VIF less than 7.5; and finally, the OLS model performance was assessed using adjusted $R^2$ [26].

*Geographically Weighted Regression (GWR)*. Unlike the OLS model, GWR is a local form of linear regression analysis that is used for modeling spatially varying relationships. GWR is usually recommended when one or more of the assumptions of the OLS model fail. Unlike the OLS model, which has one equation, GWR constructs a separate equation for each feature, which is a variable model relationship across study areas [26]. Even though the GWR model helps to explore locally varying coefficients on the map, it does not create raster coefficient layers.

## Ethical approval and consent to participate

After we submitted the research question and justification of this study to the DHS program database, we obtained a DHS dataset authorization letter from the DHS program to conduct this study. Ethical considerations for DHS data stated that it had been reviewed and approved by the ICF Institutional Review Board (IRB). Moreover, EDHS proposals were reviewed by the ICF IRB and by an IRB in Ethiopia. The ICF IRB confirmed that the survey agreed with the U. S. Department of Health and Human Services regulations for the protection of humans, while the Ethiopian IRB ensured that the survey agreed with the laws and norms of the nation.

Before each interview was performed, an informed consent statement was read to each participant, who might accept or decline to participate. The informed consent maintained voluntary participation and autonomy while strictly maintaining confidentiality and privacy. Each respondent's interview was identified only by a series of EAs, household number, and individual number. Furthermore, the geographic coordinates of each survey were displaced at a random distance and in a random direction. The displacement distance was up to two kilometers for urban EAs and up to five kilometers for rural EAs, with one percent of randomly selected

**Table 2. Socio-demographic and economic characteristics of households in Ethiopia, 2019.**

| Household characteristics | Categories | Weighted Frequencies (n) | Percentages (%) |
|---|---|---|---|
| Age of house head in years | 15–24 | 650 | 7.51 |
| | 25–34 | 2,038 | 23.52 |
| | 35–44 | 2,084 | 24.05 |
| | 45–54 | 1,515 | 17.49 |
| | ≥55 | 2,376 | 27.43 |
| Sex of household head | Male | 6,751 | 77.93 |
| | Female | 1,912 | 22.07 |
| Household educational attainment | No education | 4,120 | 47.56 |
| | Primary | 3,069 | 35.43 |
| | Secondary | 872 | 10.07 |
| | Higher | 601 | 6.94 |
| Has radio | No | 6,252 | 72.17 |
| | Yes | 2,411 | 27.83 |
| Has TV | No | 7,205 | 83.17 |
| | Yes | 1,458 | 16.83 |
| Wealth index | Poorest | 1,498 | 17.30 |
| | Poorer | 1,635 | 18.87 |
| | Middle | 1,675 | 19.34 |
| | Richer | 1,738 | 20.06 |
| | Richest | 2,117 | 24.44 |
| Tine to get water with round trip | Water on the premises | 1,805 | 20.83 |
| | 30 minutes or less | 4,727 | 54.57 |
| | More than 30 minutes | 2,131 | 24.60 |
| Source of drinking water | Unimproved | 2,710 | 31.28 |
| | Improved | 5,953 | 68.72 |

rural clusters displaced by up to ten kilometers. This protocol ensures that neither the individual nor the household can be identified [27].

## Results

### Socio-demographic and economic characteristics of participants

This study used a weighted sample of 8663 households (HHs). Out of those included, 2376 (27.43% of HH heads) were 55 years of age or older, and 78% of them were males. Nearly half (47.56%) of HH had no educational attainment. Out of 8663 interviewed HHs, 1,498 (17%) had the poorest wealth index. Over a quarter (27.83%) of HHs have a radio. One in four (24.60%) households had to obtain round-trip water collection in more than 30 minutes. Around one-third (31.28%) of HHs had unimproved sources of drinking water (Table 2).

### The prevalence of open defecation and its regional distribution

In this study, the weighted prevalence of open defecation (OD) in Ethiopia was 27.10% (95% CI: 22.85–31.79). The prevalence of OD varied from region to region, with urban-rural variation in Ethiopia. Afar and Somali regions had a higher proportion of OD (64.61% and 64.31.30%, respectively), compared with the national prevalence (27.1%), and whereas Addis Ababa (2.15%) and SNNPR (13.28%) had the lowest compared with others. Thirty-five percent of rural residents defecated openly (Table 3).

**Table 3. Percentage distribution of open defecation practices by region and place of residence in Ethiopia, 2019.**

| Variable | Categories | Has facility, n (%) | Open defecation, n (%) |
|---|---|---|---|
| Region | Tigray | 305 (52.09) | 280 (47.91) |
| | Afar | 31 (35.39) | 57 (64.61) |
| | Amhara | 1,440 (68.24) | 670 (31.76) |
| | Oromia | 2,408 (75.09) | 799 (24.91) |
| | Somali | 149 (35.69) | 269 (64.31) |
| | Benishangul-Gumuz | 79 (84.47) | 15 (15.53) |
| | SNNPR | 1,446 (86.72) | 221 (13.28) |
| | Gambela | 22 (61.99) | 13 (38.01) |
| | Harari | 20 (80.65) | 5 (19.35) |
| | Addis Ababa | 368 (97.85) | 8 (2.15) |
| | Dire Dawa | 47 (83.64) | 9 (16.36) |
| Place of residence | Urban | 2,406 (90.32) | 258 (9.68) |
| | Rural | 3,910 (65.18) | 2,089 (34.82) |
| National total prevalence | | 6,316 (72.91) | 2347 (27.09) |

## Spatial Variation of open defecation practices in Ethiopia

**Global spatial autocorrelation (Moran's I) analysis.** The spatial pattern of open defecation (OD) practices among households in Ethiopia was clustered across enumeration areas. The global spatial autocorrelation analysis depicted that there was a significant clustered pattern of OD in the enumeration areas of Ethiopia (Global Moran's $I$ = 0.45, Z-score = 9.88, p-value < 0.001). This output showed that the Z-score was positive, with a highly significant p-value that would be interpreted as 99% confidence for the clustering of OD across the enumeration in Ethiopia. The figures below depict the clustered patterns (on the right side) with high rates of OD across clusters in Ethiopia. The bright red and blue colors (to the right and left sides) indicated an increased significance level for which the likelihood of clustered patterns occurring by random chance was less than 1% (Fig 1).

**Cluster and outlier (Anselin Local Moran's *I*) analysis.** Anselin Local Moran's I analysis showed significant high-high clusters of OD were identified at Tigray, Afar, Northern Amhara, Somali, and Gambela regions, while low-low clusters of OD were observed at Addis Ababa, Dire Dawa, Harari, SNNPR, Southwest Oromia, and Benishangul Gumuz regions (Fig 2).

**Hotspots (Getis-Ord Gi\*) analysis of open defecation.** The hot spot areas of open defecation were detected in Tigray, Afar, northern, and western Amhara, Gambela, and the Somali region, whereas Addis Ababa, Dire Dawa, SNNPR, Benishangul Gumuz, and the Harari region were the cold spot areas of OD (Fig 3).

**Spatial interpolation of open defecation.** Ordinary kriging interpolation in unsampled EAs revealed that Afar, the eastern borders of Amhara, the northern Somali region, and Gambela were predicted to have a high prevalence of OD (Fig 4).

**SaTScan cluster analysis of open defecation.** Based on SaTScan cluster analysis, out of 8,652 total households, 2344 (27.1%) households defecated openly in the selected enumeration areas. A total of 150 significant clusters were detected in the nine most likely clusters.

The first most likely big cluster encompassed mainly the Tigray region, northern and eastern Amhara, and the northern Afar region. It was found at (14.100614 N, 38.304654 E)/292.36 km radius. In the primary most likely cluster, 192 (54%) HHs defected openly, and those HHs in this cluster were 2.46 times more likely to defecate openly compared with those outside this window (RR = 2.46, LLR = 272.97; P-value < 0.001) (Table 4 and Fig 5).

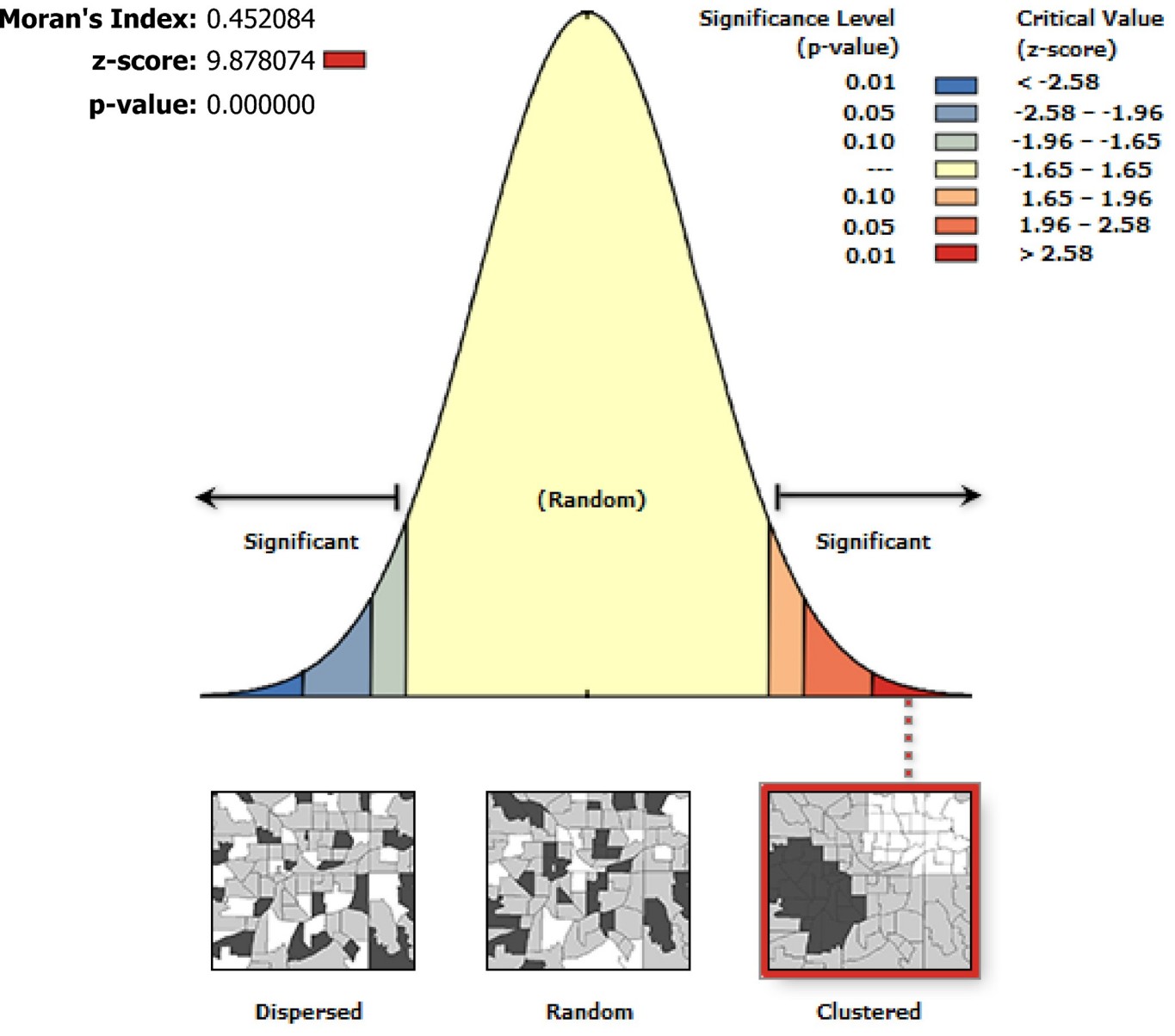

**Moran's Index:** 0.452084
**z-score:** 9.878074
**p-value:** 0.000000

Given the z-score of 9.87807429586, there is a less than 1% likelihood that this clustered pattern could be the result of random chance.

**Fig 1. Spatial patterns of open defecation among households in Ethiopia, 2019.**

The second most likely SaTScan cluster covered mainly the Somali region and the eastern and southeastern Oromia region, which was located at (6.459193 N, 42.199432 E)/315.22 km radius. Clusters in the second SaTScan window were 2.79 times more likely to have open defecation compared with those outside this window (RR = 2.79, LLR = 190.90; P-value < 0.001) (Table 4 and Fig 5). The third most likely cluster encompassed mainly the northern Oromia region. This SaTScan cluster was found at (9.531226 N, 38.081685 E)/67.38 km radius, and in

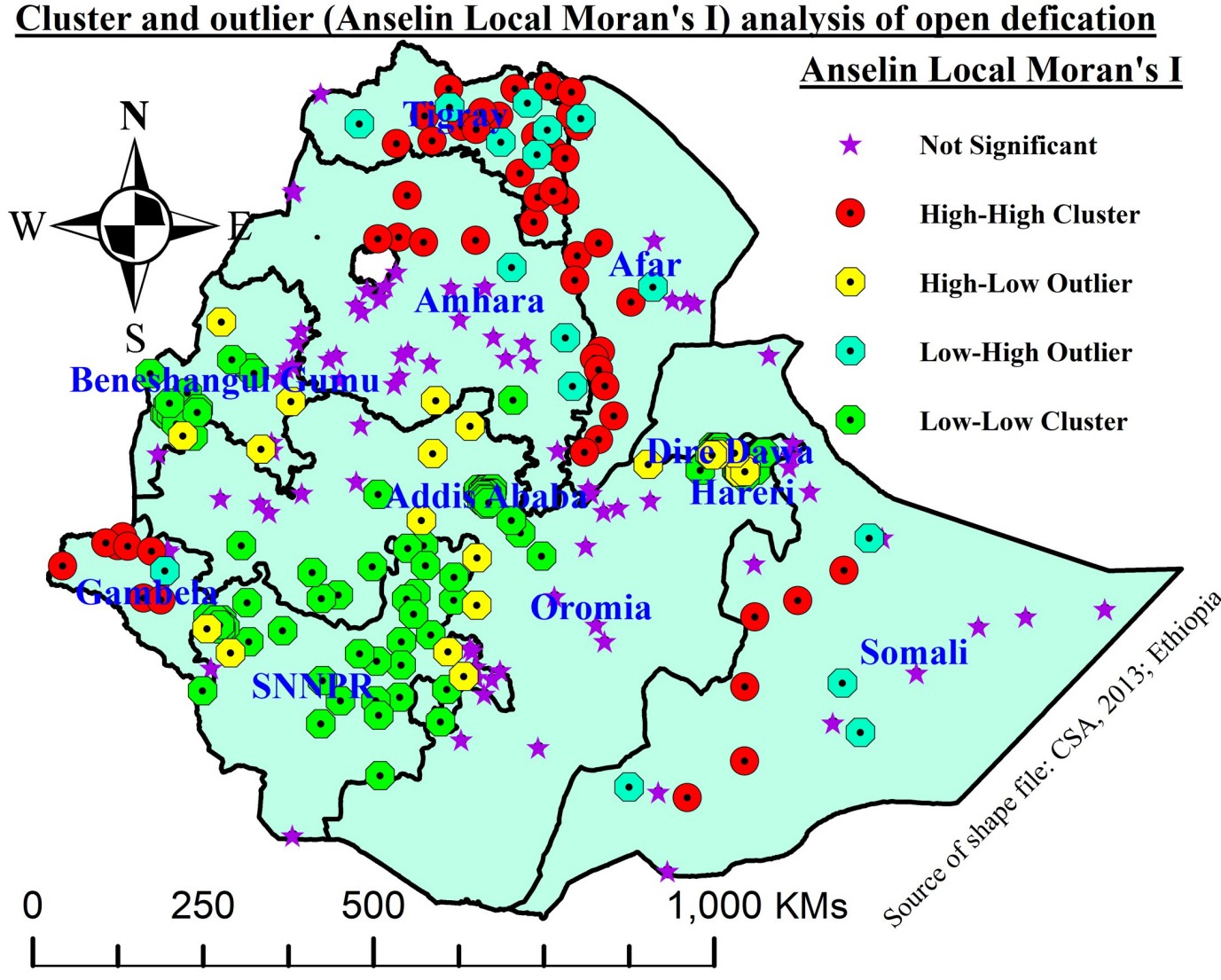

**Fig 2. Local cluster and outlier (Anselin Local Moran's *I*) analysis.**

this cluster, 115 (79.3%) HHs defecated openly. Clusters in this window were 4.70 times higher outside this window to have OD (RR = 4.70, LLR = 131.82, P-value less than 0.01) (Table 4 and Fig 5).

The fourth most likely cluster covered mainly Addis Ababa, the central and south-western parts of the Oromia region, and SNNPR. It was found at (8.651588 N, 39.118340 E)/65.24 km radius. Clusters in this window were 93% less likely to have open defecation compared with those outside this window (RR = 0.07, LLR = 93.57, P-value less than 0.001) (Table 4 and Fig 5).

The fifth most likely SaTScan window was mainly found in the southern Amhara and Afar region at (6.540286 N, 36.627468 E)/178.05 km radius. The clusters in this window were 69% lower than those outside this window for having open defecation (RR = 0.31, LLR = 89.08, P-value less than 0.001). The six most likely clusters were located at the eastern Oromia and Somali borders at (9.548779 N, 40.084216 E)/93.61 km radius. In this most likely cluster, 94 (57.3%) households were defecating openly, and clusters in this window were 2.77 times higher

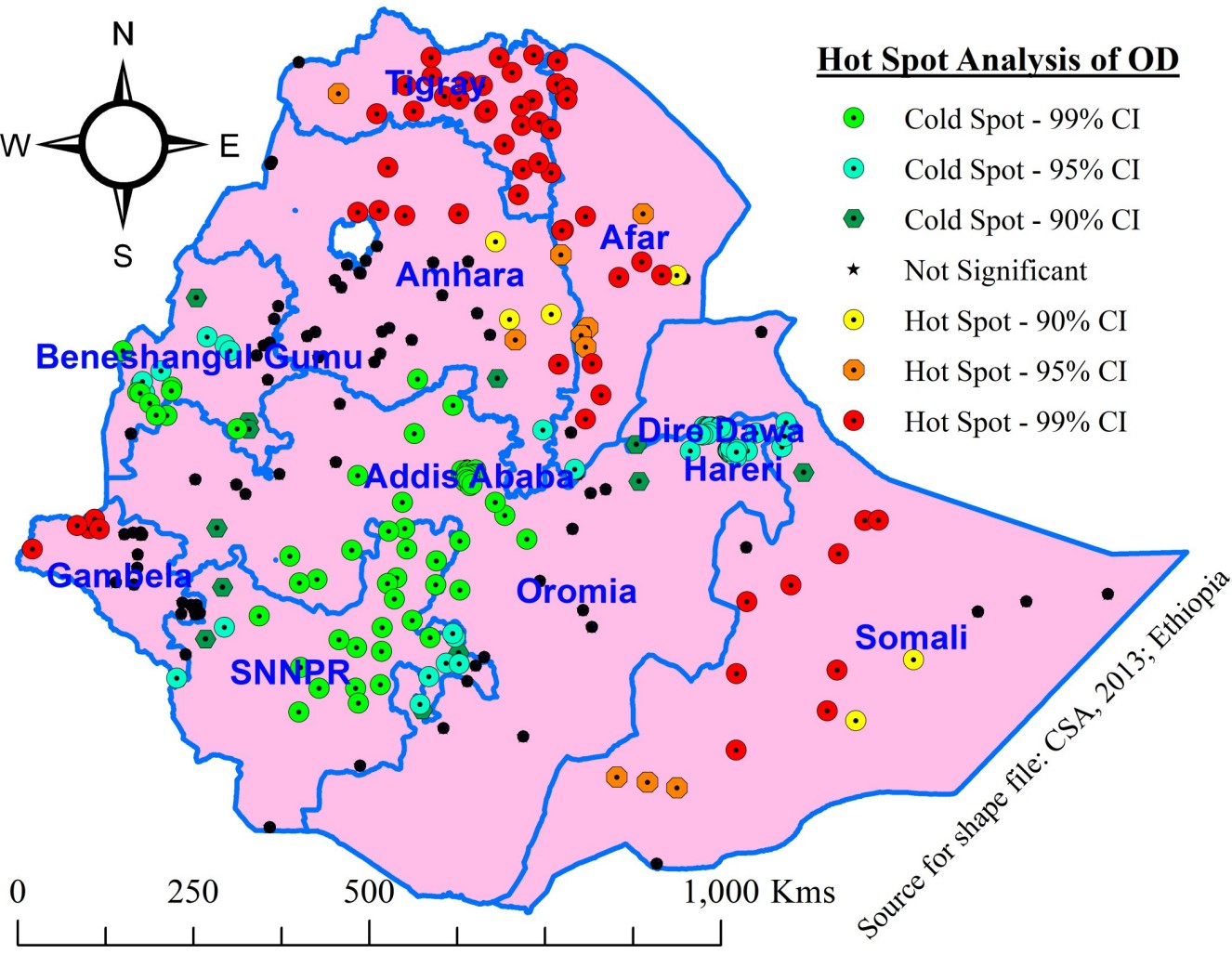

**Fig 3. Hotspot analysis (Getis-Ord Gi* statistic of open defecation among households in Ethiopia, 2019.**

than those outside this window for having open defecation (RR = 2.77, LLR = 50.07, P-value less than 0.001) (Table 4 and Fig 5).

The last most likely cluster was found mainly in the eastern Amhara region (Dessie and Kombolcha towns), and it was located at (10.992773 N, 39.299566 E)/35.07 km radius. Clusters in this window were 95% less likely to have open defecation compared with those outside this window (RR = 0.05, LLR = 36.955, P-value less than 0.001) (Table 4 and Fig 5).

## Spatial regression analysis

**Ordinary least square analysis (OLS).** Six of the OLS model assumptions were checked to evaluate whether they were best fit or biased. All the coefficients of explanatory variables had either a positive or negative relationship with the OD, and some of them were statistically significant. Redundancy or multicollinearity among explanatory variables was not observed (VIF less than 7.5) (Table 5). Residuals were not clustered in locations (Moran's I index = 0.066, Zscore = 1.51, p-value = 0.13). The OLS model performance revealed that over

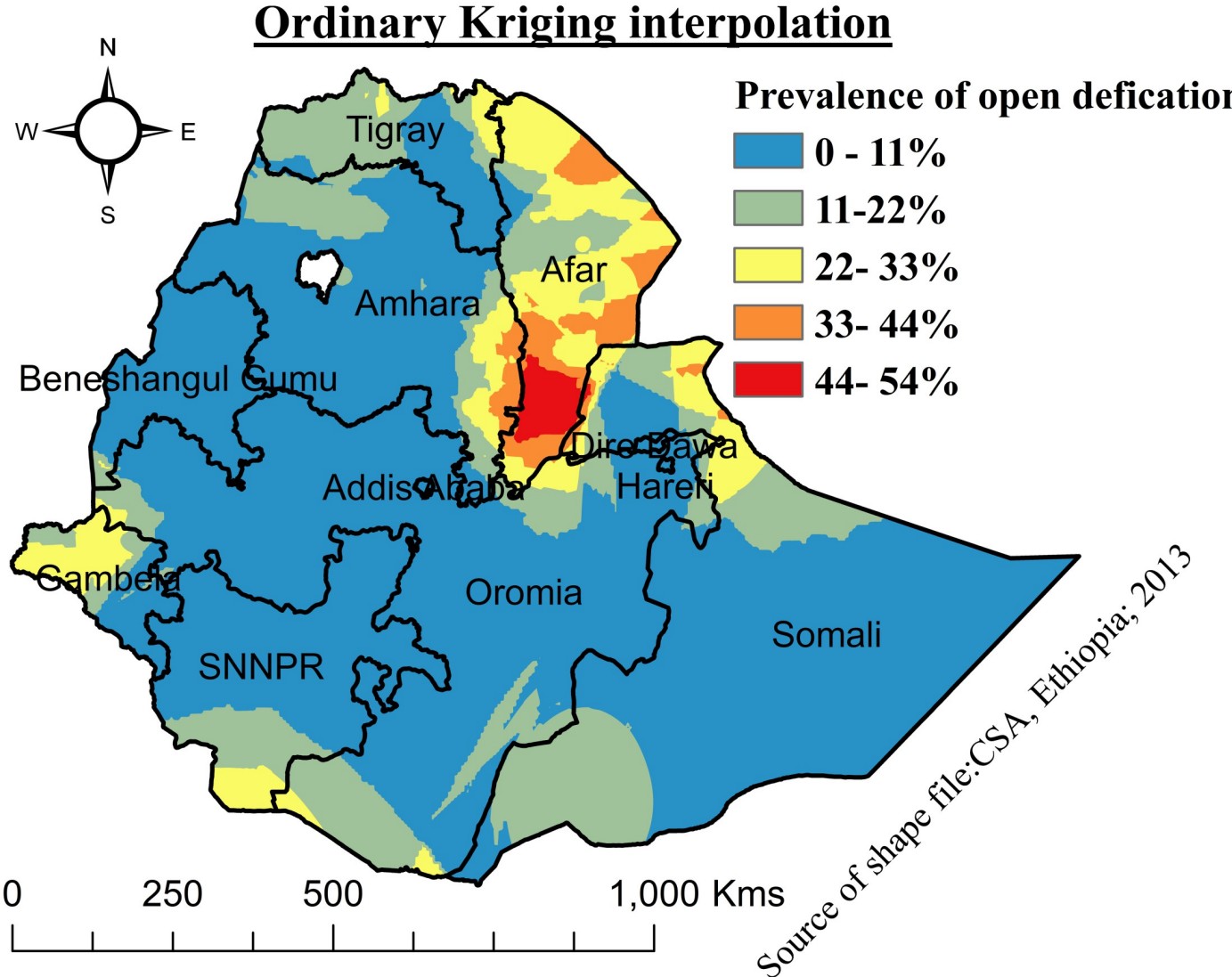

**Fig 4. Ordinary kriging interpolation of open defecation among households in Ethiopia, 2019.**

63% (adjusted R square = 0.63) of the spatial variation was explained by the model prediction, and its AICc value was -65.21.

Even though the OLS model fulfilled the above assumptions, the Jarque-Bera statistic had an asterisk or was statistically significant at a P-value < 0.05, which revealed that residuals were not normally distributed. Thus, the OLS model prediction was biased. Moreover, the Koenker (BP) statistic was statistically significant at p-value < 0.001, which showed that the relationship between the explanatory variables and the dependent variable was not consistent due to non-stationarity or heteroskedasticity that indicated the geographical variation of coefficients across clusters. Therefore, we considered fitting the GWR regression analysis because of the non-stationarity of coefficients across EAs and the biased model in the OLS regression.

Joint F and Wald statistics revealed that an overall model was significant at a p-value < 0.001. Since the Koenker (BP) statistic was statistically significant, we used robust probability to determine coefficient significance. Accordingly, the proportion of households

**Table 4. SaTScan cluster analysis of open defecation among households in Ethiopia, 2019.**

| Most likely clusters | Significant clusters detected | Coordinates/Radius | Populations | Cases n (%) | RR | LLR | P-value |
|---|---|---|---|---|---|---|---|
| 1 | 1, 6, 7, 8, 12, 13, 9, 11, 2, 14, 22, 10, 16, 17, 21, 23, 5, 56, 3, 15, 25, 35, 39, 36, 20, 4, 27, 37, 38, 24, 78, 83, 82, 18, 19, 84, 62, 55, 57, 85, 58, 61, 59, 46, 29, 45, 74 | (14.100614 N, 38.304654 E) / 292.36 km | 1388 | 192 (54) | 2.46 | 272.97 | <0.001 |
| 2 | 136, 134, 142, 138, 123, 145, 133, 137, 141, 125, 111, 131, 143, 110, 135, 122, 144, 132, 129, 106, 103, 250, 102, 248, 249, 114, 244, 247, 255 | (6.459193 N, 42.199432 E) / 315.22 km | 819 | 417 (50.9) | 2.79 | 190.90 | <0.001 |
| 3 | 99, 100 | (9.531226 N, 38.081685 E) / 67.38 km | 145 | 115 (79.3) | 4.70 | 131.82 | <0.001 |
| 4 | 101, 90, 280, 278, 279, 272, 271, 277, 274, 270, 268, 273, 269, 275, 264, 276, 267, 263, 266, 265, 260, 261, 256, 258, 257, 259, 262 | (8.651588 N, 39.118340 E) / 65.24 km | 637 | 8 (1.3) | 0.07 | 93.57 | <0.001 |
| 5 | 196, 173, 192, 204, 198, 191, 195, 199, 197, 190, 96, 201, 91, 189, 200, 194, 97, 223, 202, 215, 210, 180, 222, 224, 227, 179, 178, 221, 216, 172, 226 | (6.540286 N, 36.627468 E) / 178.05 km | 1263 | 86 (6.8) | 0.31 | 89.08 | <0.001 |
| 6 | 42, 40, 69, 41, 28, 43, 105, 127 | (9.548779 N, 40.084216 E) / 93.61 km | 164 | 94 (57.3) | 2.77 | 50.07 | 0.001 |
| 7 | 128, 130 | (9.673818 N, 42.836549 E) / 19.66 km | 27 | 27 (100) | 4.96 | 42.917 | 0.001 |
| 8 | 193 | (4.495034 N, 36.230625 E) / 0 km | 25 | 25 (100) | 5.08 | 40.35 | 0.001 |
| 9 | 63, 66, 51 | (10.992773 N, 39.299566 E) / 35.07 km | 202 | 2 (1) | 0.05 | 36.955 | 0.001 |

with rural residence (p-value < 0.01), the proportion of households with no radio (p-value <0.001), the proportion of households with a female household head (p-value <0.05), the proportion of households with the poorest wealth index (p-value <0.001), and the proportion of households with no educational attainment (p-value <0.05) were significantly associated with open defecation in Ethiopia (Table 5).

**Geographically Weighted Regression (GWR) analysis.** The local spatial regression (GWR analysis) was conducted because two of the six assumptions in the OLS regression were not met, in which both the Jarque-Bera statistic and the Koenker (BP) were significant. In the GWR analysis, model performance was improved over the OLS model regression. The adjusted R square increased from 63.16% in the OLS analysis to 75.20% in the GWR analysis, indicating that the GWR model better explained the geographical variation of OD among households in Ethiopia. Moreover, the AICc value of -65.21 in the OLS model was reduced to -128.34 in the GWR model, which revealed a better model fit to explore open defecation using GWR (Table 6).

Based on the figures below, the red color depicts the highest coefficients in each explanatory variable that showed the highest influence on the prevalence of OD among households in Ethiopia.

Accordingly, as the percentage of rural residents increased, the prevalence of OD also increased. Areas with the highest influence of OD by rural residency were observed mainly in Tigray, Afar, Amhara, Benishangul Gumuz, and the Northern Oromia regions, while areas with the lowest effects were found in Addis Ababa, SNNPR, Gambela, and the Somali regions (Fig 6).

Similarly, as the percentage of female households increased, the prevalence of open defecation also increased. Areas with the highest percentages of female HH heads were found in Western Tigray, Afar, western Amhara, western SNNPR, Gambela, eastern Oromia, Dire-Dawa, Harari, and Somali regions; in contrast, eastern Tigray, southern and eastern Amhara, central and western Oromia, northern Benishangul Gumuz, and eastern and northern SNNPR were depicted to have the lowest coefficients and influence on OD (Fig 7).

## SaTScan cluster analysis of high and low rates of OD in Ethiopia, 2019

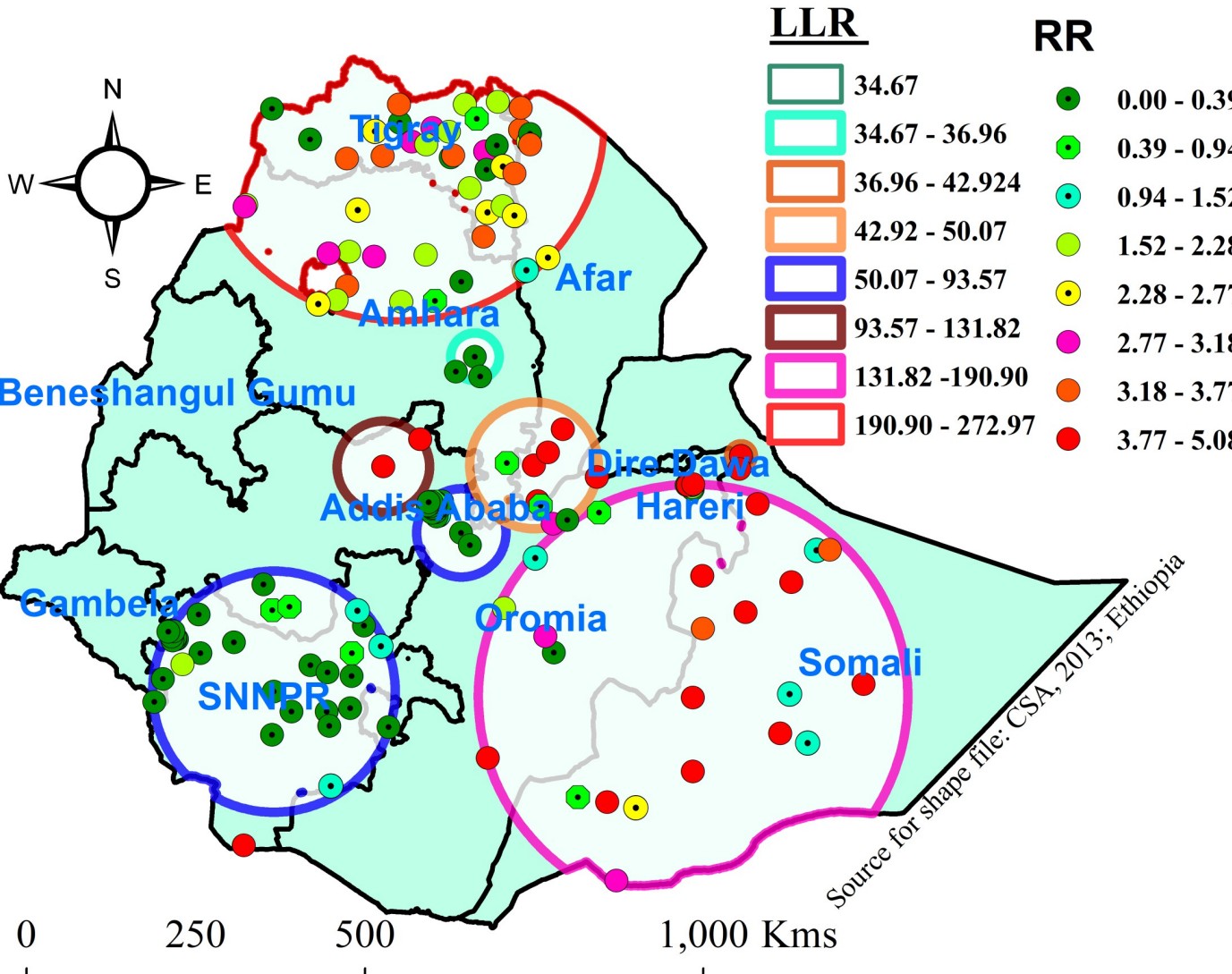

**Fig 5. SaTScan cluster analysis of open defecation among households in Ethiopia, 2019.**

This study showed that as the percentage of households with no educational attainment increased, the prevalence of open defecation also increased. Areas with the highest coefficients of HHs with no educational attainment were observed in the regions of Tigray, Afar, Amhara, Gambela, Eastern Oromia, and Northern Somali, whereas areas with the lowest coefficients were explored in Addis Ababa, SNNPR, Benishangul Gumuz, western Oromia, and southern Somali (Fig 8).

Likewise, as the percentage of households with no radio increased, the prevalence of open defecation also increased. Coefficients with the highest percentage of HHs with no radio were observed in the western Tigray, Afar, western Amhara, Gambela, eastern Oromia, Dire Dawa, Harari, and Somali regions, while coefficients with the lowest percentage of HHs with no radio were shown in the regions of eastern and southern Tigray, eastern and southern Amhara,

**Table 5. Summary and model diagnostic outputs of OLS model fitting results for open defecation among households in Ethiopia, 2019.**

| Explanatory variables | Coefficients | Standard error | t-statistic | Probability | Robust SE | Robust-pr | VIF |
|---|---|---|---|---|---|---|---|
| Intercept | -0.18 | 0.06 | -3.01 | 0.003* | 0.06 | 0.002* | - - - - |
| Proportion of rural residents | 0.12 | 0.04 | 3.21 | 0.002* | 0.04 | 0.001* | 2.02 |
| Proportion of HH head ≥55 years | -0.14 | 0.10 | -1.40 | 0.163 | 0.10 | 0.175 | 1.19 |
| Proportions of no radio in the HHs | 0.25 | 0.08 | 3.12 | 0.002* | 0.07 | 0.001* | 1.68 |
| Proportion of female HH head | 0.24 | 0.08 | 3.02 | 0.003* | 0.09 | 0.011* | 1.33 |
| Proportions of the poorest HHs | 0.55 | 0.05 | 10.38 | < 0.001* | 0.06 | < 0.001* | 2.02 |
| Proportions of HHs with no educational attainment | 0.18 | 0.08 | 2.32 | 0.02* | 0.08 | 0.016* | 2.42 |

| Ordinary Least Square regression model diagnostic results | | | | |
|---|---|---|---|---|
| Number of observations | 305 | Corrected Akaike's Information Criterion (AICc) | | -65.21 |
| Joint F-Statistic | 87.86 | Adjusted R-Squared | | 63.2% |
| Joint Wald Statistic | 967.79 | Prob(>F), (6,298) degrees of freedom | | < 0.001* |
| Koenker (BP) Statistic | 44.74 | Prob(>chi-squared), degrees of freedom: | | < 0.001* |
| Jarque-Bera Statistic | 6.14 | Prob(>chi-squared), degrees of freedom: | | < 0.001* |
| | | Prob(>chi-squared), (2) degrees of freedom: | | 0.046* |

* = Significant explanatory variables at p-value less than or equal to 0.05

**Table 6. Summary of geographically weighted regression (GWR) analysis of open defecation among households in Ethiopia, 2019.**

| Neighbors | 173 | AICc | -128.34 |
|---|---|---|---|
| Effective number | 38.20 | Adjusted R square | 72% |
| Sigma | 0.19 | | |

Addis Ababa, Benishangul Gumuz region, central and western Oromia, and northern and eastern SNNPR (Fig 9).

In this study, as the percentage of the poorest households increased, the prevalence of open defecation also increased. Coefficients with the highest percentage of the poorest households were observed in the western Tigray, southern Afar, Amhara (northwestern, south Wolo, and north Shewa), Gambela, western SNNPR, Addis Ababa, eastern Oromia, Dire Dawa, Harari, and Somali regions, while coefficients with the lowest percentage of the poorest households were observed in the regions of eastern and southern Tigray, northern Afar, eastern, central, and southern Amhara, Benishangul Gumuz region, northern and eastern SNNPR, and eastern Somali region (Fig 10).

## Discussion

Existing evidence indicates that people often prefer open defecation (OD) practices rather than using designated toilets in Ethiopia. A recent study showed that over one in four (27.8%) households practiced OD despite the presence of latrine facilities in Ethiopia [21], and around 1 out of 6 (15.9%) households practiced OD after they had been certified open defecation-free status [24]. Therefore, this study aimed to explore the spatial variation and geographical inequalities of OD among households in Ethiopia using recent national survey data.

In this study, over a quarter of total households or 2,347 households, (27.10%, 95% CI: 22.85–31.79), defecated openly in Ethiopia in 2019. This finding was comparable with a study in Nigeria (25.1%) [22] and Haiti (25.3%) [28].

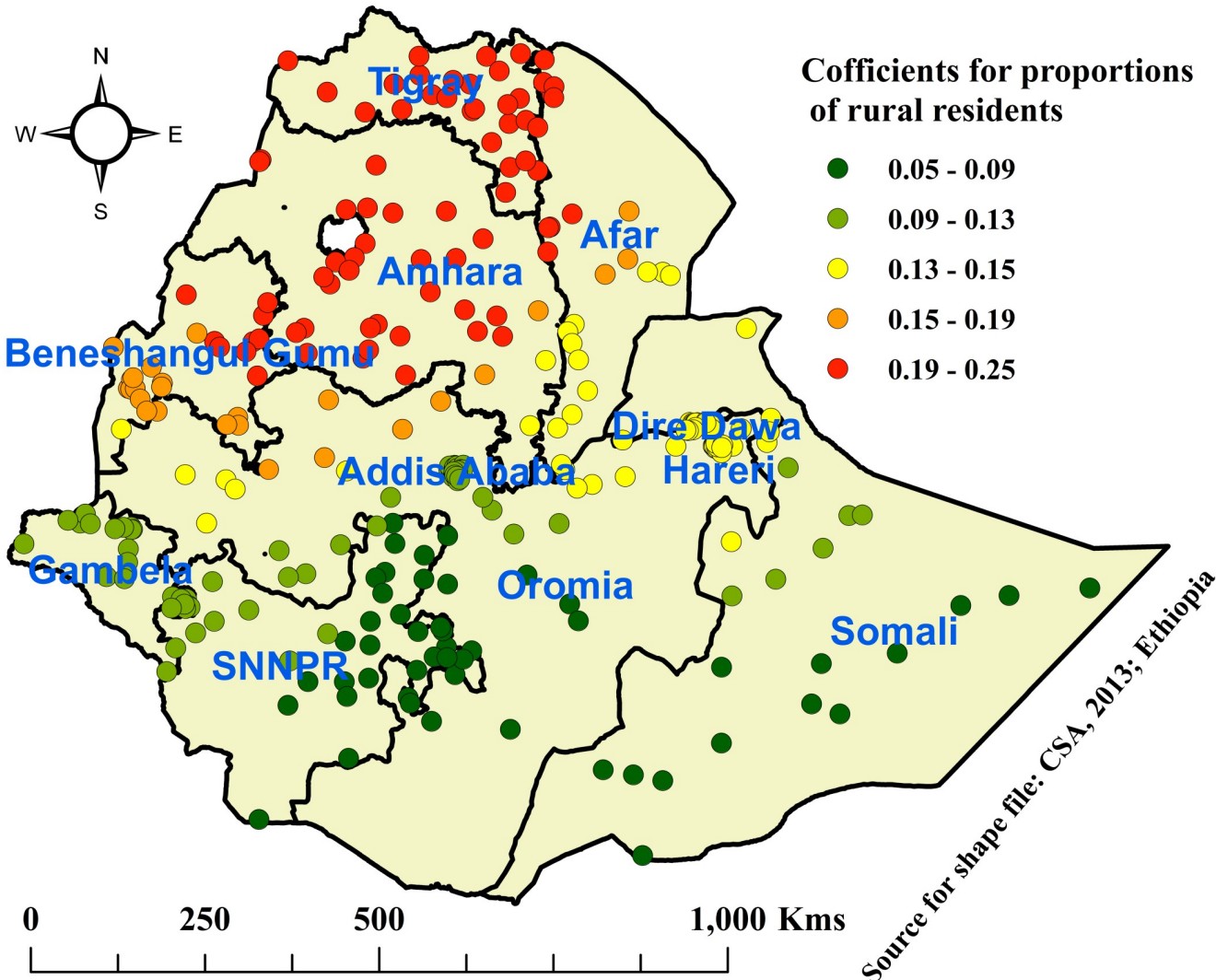

**Fig 6. Coefficients for the proportion of rural residence associated with open defecation among households in Ethiopia, 2019.**

However, the prevalence of this study was lower than studies from Ghana (44.2% and 42%) [13, 29], the residential zones of Akure, Nigeria (34.2%) [30], and Kaduna, Nigeria (35%) [31]. The variation could be because the Ethiopian government adopted and practiced urban and rural health extension packages, which aid in an increased number of latrine constructions per household with basic sanitation [32], reducing the burden of OD from 62.2% [33] in 2005 to 27% in 2019 [15]. However, the current prevalence of OD is still the highest in eastern Africa, and Ethiopia is not on the best track to achieve the SDG goal of being open defecation-free in 2030. On the other hand, this study was higher than studies conducted in Kenya (10%) [34], a recent study in Haiti (20%) [5], in south Asian countries such as India (20%) [7, 8], Bangladesh (1.2%) [9], Nepal (7%) [35], and Pakistan (16%) [36]. The possible variation could be because of cultural influence, socio-economic variations like poverty, weak rules and regulations, and the absence of monitoring for sustainability after a designated latrine.

The findings of this study revealed that OD had spatial variation across enumeration areas (EAs) in Ethiopia. The global spatial autocreation showed that OD was clustered across EAs.

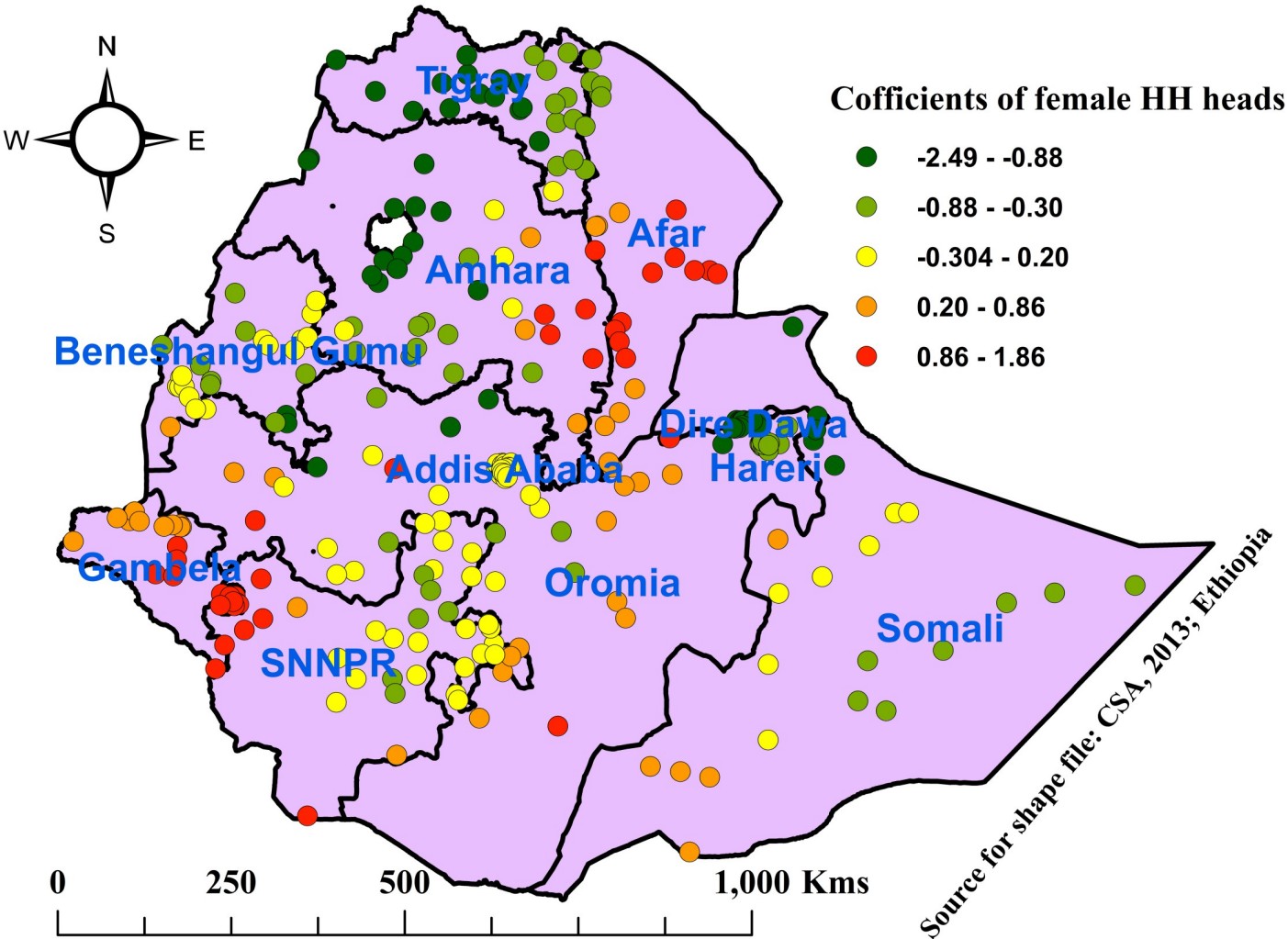

**Fig 7. Coefficients for proportions of female household heads associated with open defecation among households in Ethiopia, 2019.**

However, it does not show the specific places of high- or low-level clusters. Therefore, the local spatial autocorrelation, or Anselin Local Moran's I, in this study revealed that Tigray, Afar, Northern Amhara, Somali, and Gambela regions were found to have high clusters for OD, while Addis Ababa, Dire Dawa, Harari, SNNPR, Southwest Oromia, and Benishangul Gumuz regions were observed to have low clusters for OD. Likewise, ordinary kriging interpolation in unobserved EAs also showed that the Afar region, the eastern borders of Amhara, the northern Somali region, and Gambela were predicted to have a high prevalence of OD. This was mainly because nomadic way of life standards, high rural-urban disparity, poverty-related factors, and poor health extension package (HEP) implementation, monitoring, and evaluation in the highly concentrated areas. This was also supported by a study conducted in India [7, 8]. Recent evidence showed that households that owned the designed toilets preferred OD due to many reasons, like cultural and attitude-related factors, a lack of awareness, and having a large family size in the above areas [37].

Similarly, the hotspot areas with OD were detected mainly in the five regions of Ethiopia, namely, Tigray, Afar, Amhara (northern and western part), Gambela, and the Somali regions,

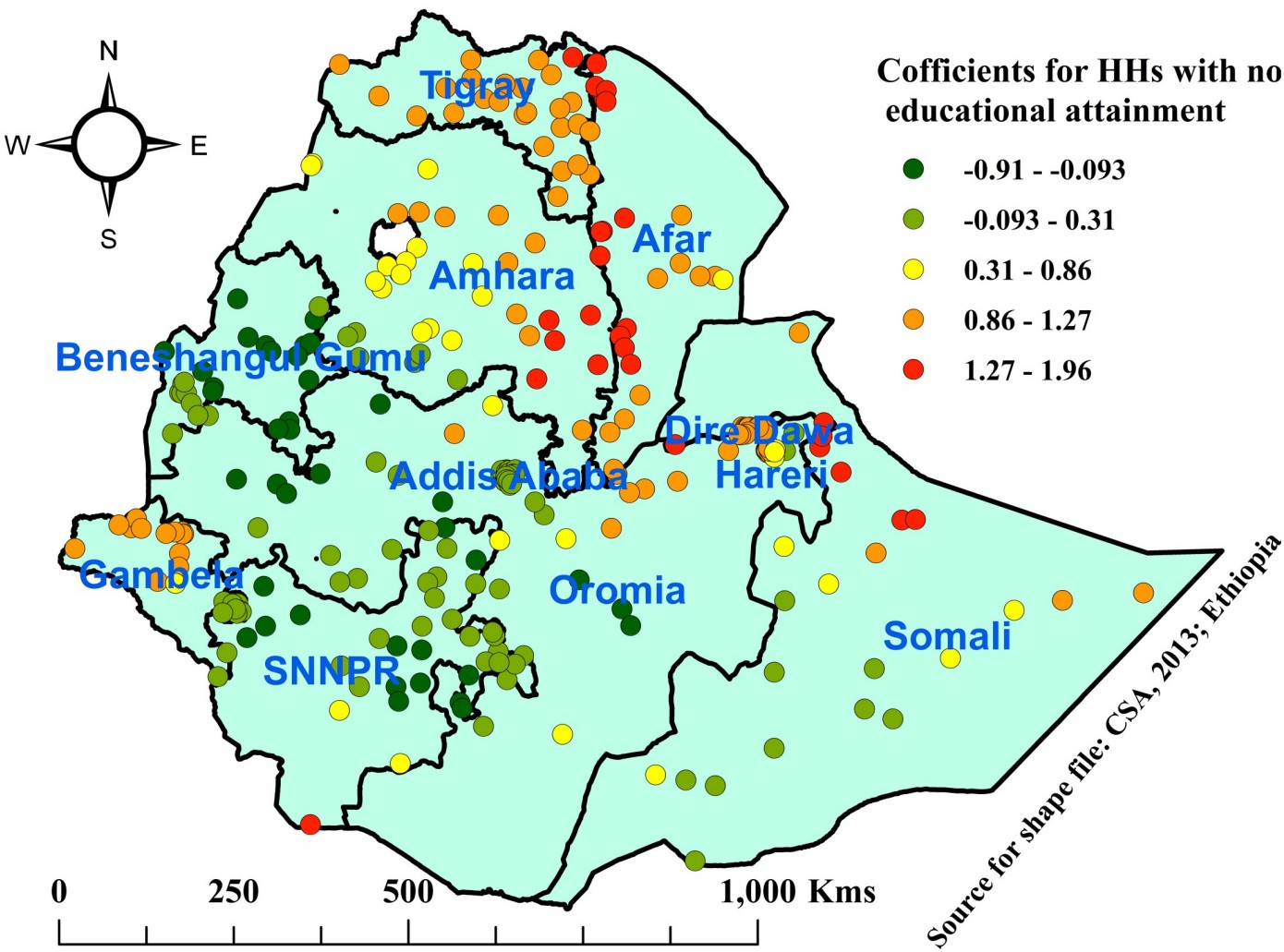

**Fig 8. Coefficients for proportions no educational attainment among households associated with open defecation in Ethiopia, 2019.**

while Addis Ababa, central and western Oromia, Benishangul Gumuz, SNNPR, Dire Dawa, and Harari were explored to have cold spots. This might be because OD varied from region to region (65% in Afar vs. 2% in Addis Ababa) with urban to rural variations (10% vs. 35% in Ethiopia). Moreover, the variation in OD might be attributed to a lack of income, culture, especially among rural residents, and the nomadic way of life in northern and eastern Ethiopia. On the other hand, local studies conducted in central Oromia [38] and SNNPR [39, 40] revealed that half of HHs utilized the designated toilet facilities, and this might be because of improved and sustained HEPs in central and southern Ethiopia [41].

In this study, cluster scan analysis explored high and low rates of OD among households in which more than half of HHs utilized the designated toilet facilities, and this might be because of improved and sustained HEP in central and southern Ethiopia [41]. Ethiopia. The first most likely cluster encompassed mainly the northern parts of Ethiopia, namely the Tigray, Afar, and Amhara regions, and more than half (54%) of households (HHs) practiced OD in these areas. This most likely big cluster revealed that households in this cluster were 2.46 times more likely to defecate openly compared with those outside this window. Similarly, the second most likely

### Propotions of households with no radio associated with open defecation in Ethiopia, 2019

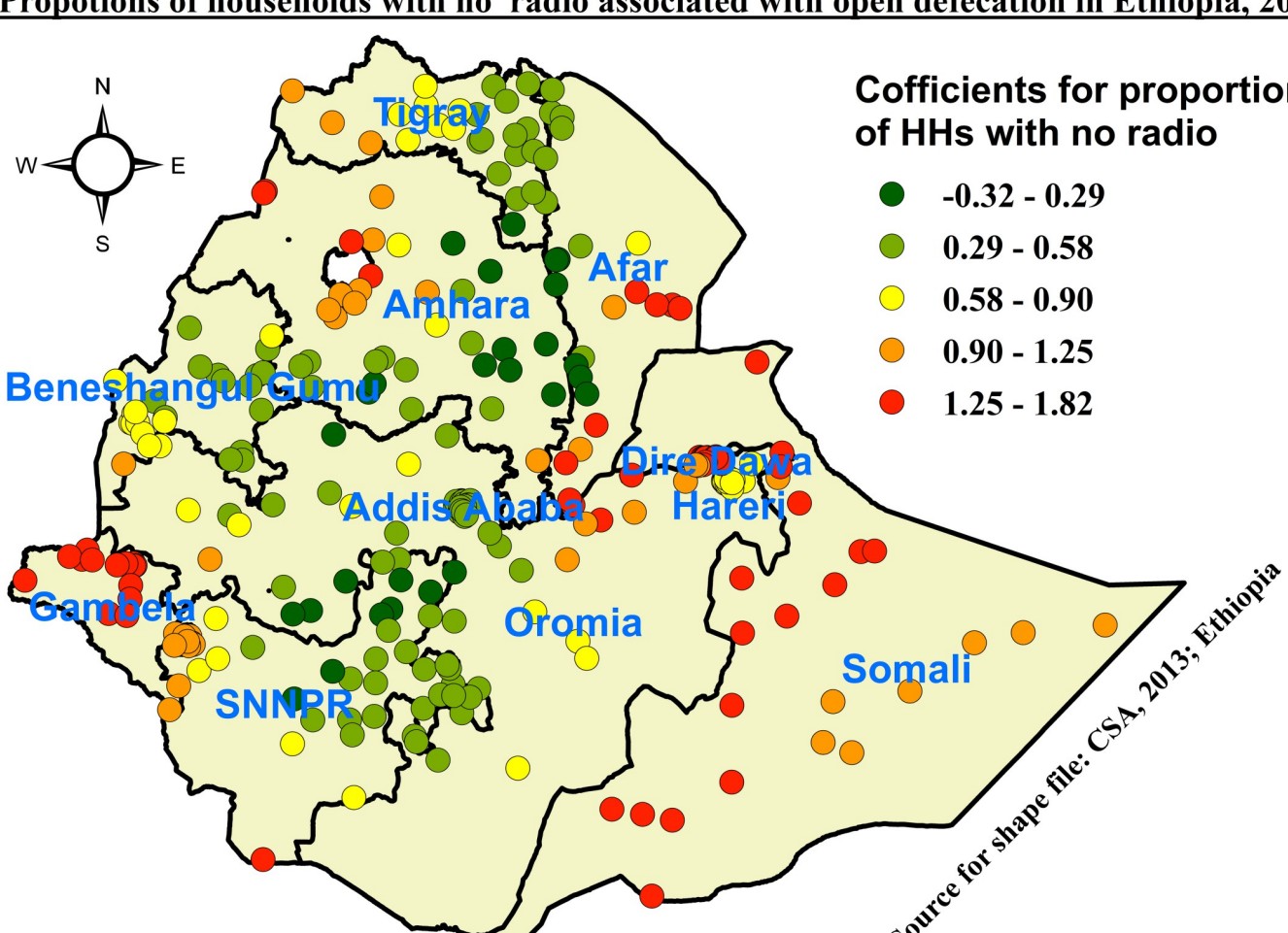

**Fig 9. Coefficient for no radio in the household associated with open defecation among households in Ethiopia, 2019.**

SaTScan cluster was found in eastern Oromia and mainly in the Somali region. Clusters in the second window had OD practices that were 79% higher than those outside this window. The third most likely cluster encompassed mainly the northern Oromia region, and 79% of HHs practiced OD in the enumeration areas (EAs). Clusters in this window were 4.70 times more likely than clusters outside this window to have OD. The fourth most likely cluster encompassed mainly the Addis Ababa and SNNPR regions. Clusters in this window were 93% less likely to have open defecation compared with those outside this window. The fifth most likely SaTScan window was mainly found on the southern border of the Amhara and Afar regions, and the clusters in this window were 69% lower than those outside this window for having OD practice. The six most likely clusters were located at the border of the eastern Oromia and Somali regions, and over half (57%) of households practiced OD in this area. Clusters in this window were 77% higher than those outside this window for having open defecation. The last most likely cluster was found mainly in the eastern Amhara region (Dessie and Kombolcha towns), and clusters in this window were 95% less likely to have OD compared with those

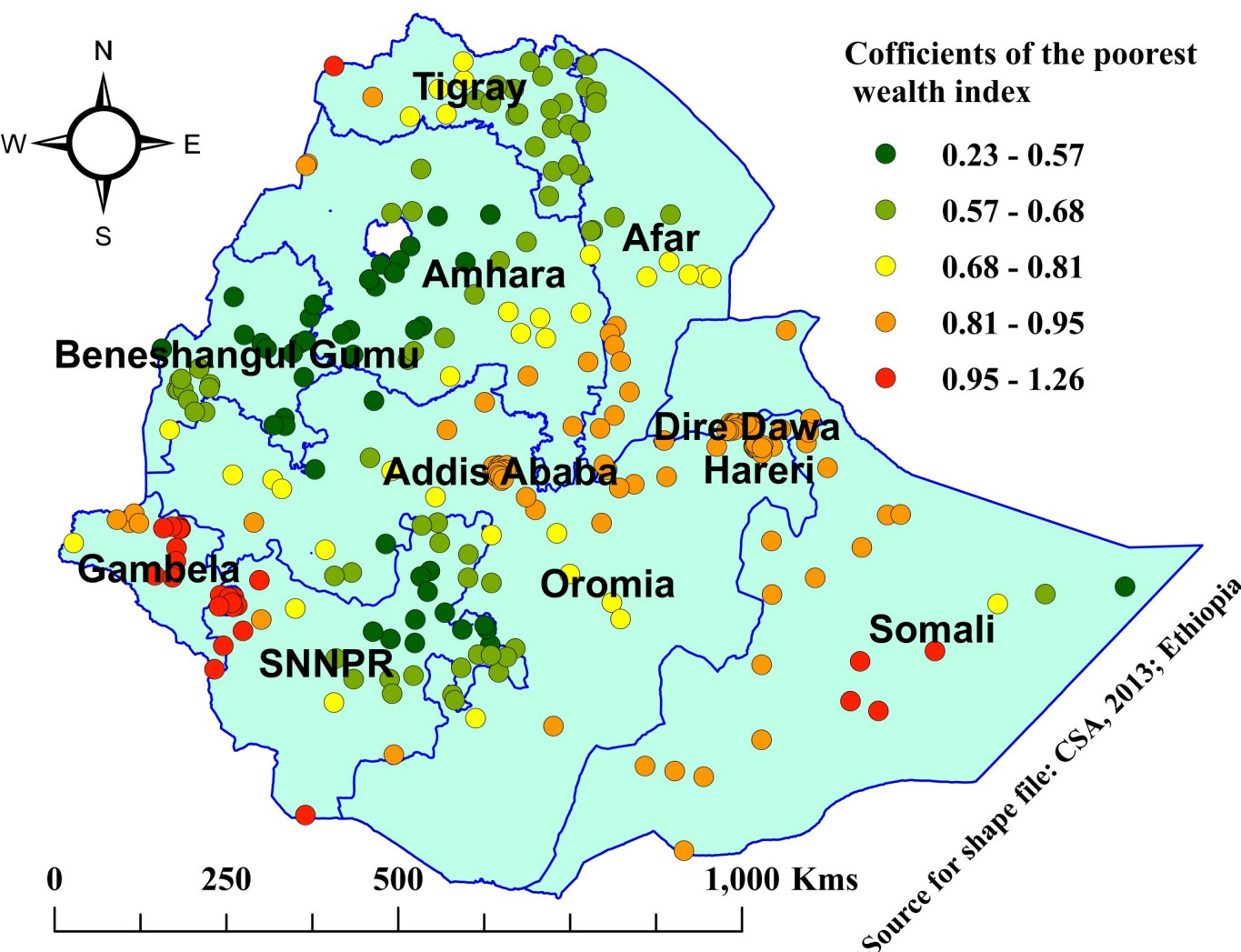

**Fig 10. Coefficients for proportions of the poorest households associated with open defecation among households in Ethiopia, 2019.**

outside this window. The spatial variation of OD in northern Ethiopia could be attributed to poor implementation of HEP because of the challenging topography, presence of weak monitoring and evaluation systems for HEPs, and the nomadic way of life in north-eastern Ethiopia, which might tend to have OD. This was supported by a study in Kenya [42].

In this study, the spatial regression analysis revealed that sociodemographic and economic factors affected OD practice with spatial non-stationarity across EAs in Ethiopia. The GWR finding showed that there was a positive association between rural residence and the practice of OD. Accordingly, as the percentage of rural residents in the EAs was high, the prevalence of OD also increased. This was supported by similar studies conducted in India [7, 8]. Regions such as Tigray, Afar, Amhara, Benishangul Gumuz, and the northern Oromia regions were explored to have a higher influence of rural residency on OD, while regions with a lower effect of rural residency on OD were found in Addis Ababa, SNNPR, Gambela, and the Somali regions. This might be rated with a high proportion of poverty, low education coverage, cultural influence, and a poor attitude among the rural residents toward sustained use of latrines. Rural residents have increased odds of constructing latrine facilities and even despite the

presence of latrine facilities, and even despite the presence of latrine facilities, they prefer defecating in the bushes, fields, or nearby rivers. This is mainly because of cultural influence, lack of financial sources, poor attention, and reluctance. It was supported by studies in Kenya (10%) they prefer defecating in the bushes or fields or nearby rivers. This is mainly because cultural influence, lack of financial source, poor attention, and being reluctance. It was supported by studies in Kenya (10%) [34], Nigeria [22, 43], and India [44].

Similarly, as the percentage of households with no educational attainment increased in specified areas, the prevalence of OD also increased in those areas. Regions of Tigray, Afar, Amhara, Gambela, Eastern Oromia, and Northern Somali were observed with the highest proportions of households with no educational attainment, which influence the practice of OD, whereas areas with the lowest influence by proportions of no educational attainment were explored in Addis Ababa, SNNPR, Benishangul Gumuz, western Oromia, and southern Somali. The possible variation could be cultural influence, limited knowledge, and awareness about the consequences of OD among people with no education, and poor monitoring and regulation in those areas, especially poor implementations of health extension. This finding was supported by a study in Kenya [34, 42], Nigeria [43], Haiti [28], and India [7, 8], which revealed OD was associated with no education among households.

The findings of this study also revealed that as the percentage of female household heads increased, the prevalence of OD also increased across study areas. Areas with the highest percentages of female household heads that were positively associated with OD were detected similarly in western Tigray, Afar, western Amhara, western SNNPR, Gambela, eastern Oromia, Dire-Dawa, Harari, and Somali regions, while eastern Tigray, southern and eastern Amhara, central and western Oromia, northern Benishangul Gumuz, and eastern and northern SNNPR were depicted to have the lowest proportions of female household heads with lower effects on OD. This might be related to a lack of income and place for latrine construction, poorly constructed latrine, and inconsistent use because it is not frequently cleaned. This finding was supported by a study in Haiti [28]. In contrast to this, male HH heads were associated with OD, as reported by a study in Nigeria [43].

Likewise, households that did not listen to the radio were positively associated with the practice of OD. In this study, as the percentage of households with no radio increased in specified communities, the prevalence of OD also increased in those EAs. The highest percentage of households with no radio were observed in the western Tigray, Afar, western Amhara, Gambela, eastern Oromia, Dire Dawa, Harari, and Somali regions, while areas with the lowest percentage of households with no radio that had less effect on OD were explored in the regions of eastern and southern Tigray, eastern and southern Amhara, Addis Ababa, Benishangul Gumuz region, central and western Oromia, and northern and eastern SNNPR. This might be because people who often listen to the radio get key messages and awareness about the harmful effects of OD. Awareness creation and health education via radio could change people's behavior toward basic sanitation and their hatred of OD. In some parts of the country, people often have a workload to listen to the radio. This was supported by similar studies in Haiti [28].

The findings of this study also revealed that as the percentages of the poorest households increased in specified communities, the prevalence of OD also increased in those EAs. The highest percentage of the poorest households that influence OD were observed in the western Tigray, southern Afar, Amhara (northwestern, south Wolo, and north Shewa), Gambela, western SNNPR, Addis Ababa, eastern Oromia, Dire Dawa, Harari, and Somali region, while the lowest percentage of the poorest households, which has a lower effect on OD, were observed in the regions of eastern and southern Tigray, northern Afar, eastern, central, and southern Amhara, Benishangul Gumuz region, northern and eastern SNNPR, and eastern Somali region. This is because wealth is a proxy indicator for improved latrine construction and sustainable basic

sanitation. Including in large cities, poor households cannot afford to construct improved latrines; as a result, they prefer OD. This finding was supported by studies in Kenya that reported poverty was the major contributing factor for OD [34, 42], Nigeria [43], and Haiti [28].

## Strengths and limitations

We conducted a countrywide, nationally representative study to explore geospatial variation and its determinants of open defecation in Ethiopia that aid in designing interventions locally to achieve SDG goals in 2030. However, since this study used survey data, another important variable was not included such as cultural influence or other individual-level variables like behavioral factors like attitude-related characteristics that would determine open defecation.

## Conclusion and recommendations

In this study, the prevalence (27.10%) of OD is high enough to achieve the SDG goal of being open defecation-free by 2030. The spatial pattern of OD in Ethiopia was clustered across EAs. Clusters with high OD practices were concentrated in northern, eastern, and western Ethiopia. Similarly, the hotspot areas of OD were mainly detected in northern Ethiopia, namely Tigray, Afar, Amhara (northern and western part), Gambela, and the Somali region, while Addis Ababa, central and western Oromia, Benishangul Gumuz, SNNPR, Dire Dawa, and Harari were cold spots areas of OD. Local cluster scan analysis revealed that high-risk most likely clusters were observed mainly in the northern, eastern, and south-eastern parts of Ethiopia, whereas central and southern Ethiopia were detected to have low risk for OD. Being a rural resident, female household head; HHs with no educational attainment, HHs with no radio, and having the poorest wealth index positively influenced OD practice.

Therefore, the government of Ethiopia and stakeholders need to design interventions in high clusters, hot spot areas, and high-risk local clusters identified in northern, eastern, and western parts. The program managers should plan interventions and strategies like encouraging health extension programs, which aid in facilitating basic sanitation facilities in rural areas and the poorest HHs, including encouraging female HHs, as well as community mobilization with awareness creation, especially for those who are uneducated and who do not have radios.

## Acknowledgments

Our deepest-rooted gratitude goes to the DHS program, which granted us permission to use DHS datasets to conduct this study. We would also like to thank the Ethiopian Public Health Institute and stakeholders for their contribution to collecting and organizing the second EMDHS dataset.

## Author Contributions

**Conceptualization:** Nebiyu Mekonnen Derseh.

**Data curation:** Nebiyu Mekonnen Derseh, Muluken Chanie Agimas, Tigabu Kidie Tesfie, Habtamu Wagnew Abuhay.

**Formal analysis:** Nebiyu Mekonnen Derseh, Meron Asmamaw Alemayehu, Muluken Chanie Agimas, Getaneh Awoke Yismaw, Tigabu Kidie Tesfie, Habtamu Wagnew Abuhay.

**Funding acquisition:** Getaneh Awoke Yismaw, Tigabu Kidie Tesfie.

**Investigation:** Nebiyu Mekonnen Derseh, Meron Asmamaw Alemayehu, Muluken Chanie Agimas, Getaneh Awoke Yismaw, Tigabu Kidie Tesfie, Habtamu Wagnew Abuhay.

**Methodology:** Nebiyu Mekonnen Derseh, Meron Asmamaw Alemayehu, Muluken Chanie Agimas, Getaneh Awoke Yismaw, Tigabu Kidie Tesfie, Habtamu Wagnew Abuhay.

**Project administration:** Nebiyu Mekonnen Derseh, Meron Asmamaw Alemayehu, Muluken Chanie Agimas, Getaneh Awoke Yismaw, Tigabu Kidie Tesfie, Habtamu Wagnew Abuhay.

**Resources:** Nebiyu Mekonnen Derseh, Meron Asmamaw Alemayehu, Muluken Chanie Agimas, Getaneh Awoke Yismaw, Tigabu Kidie Tesfie, Habtamu Wagnew Abuhay.

**Software:** Nebiyu Mekonnen Derseh, Muluken Chanie Agimas, Getaneh Awoke Yismaw, Tigabu Kidie Tesfie, Habtamu Wagnew Abuhay.

**Supervision:** Nebiyu Mekonnen Derseh, Meron Asmamaw Alemayehu, Muluken Chanie Agimas, Getaneh Awoke Yismaw, Tigabu Kidie Tesfie, Habtamu Wagnew Abuhay.

**Validation:** Nebiyu Mekonnen Derseh, Muluken Chanie Agimas, Getaneh Awoke Yismaw, Tigabu Kidie Tesfie, Habtamu Wagnew Abuhay.

**Visualization:** Nebiyu Mekonnen Derseh, Meron Asmamaw Alemayehu, Muluken Chanie Agimas, Getaneh Awoke Yismaw, Tigabu Kidie Tesfie, Habtamu Wagnew Abuhay.

**Writing – original draft:** Nebiyu Mekonnen Derseh, Meron Asmamaw Alemayehu, Muluken Chanie Agimas, Getaneh Awoke Yismaw, Tigabu Kidie Tesfie, Habtamu Wagnew Abuhay.

**Writing – review & editing:** Nebiyu Mekonnen Derseh.

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
