## [Decision Letter · Decision Letter 0]

10 Apr 2024

PONE-D-24-01346Spatial variation and geographical weighted regression analysis to explore open defecation practice and its determinants among households in EthiopiaPLOS ONE

Dear Dr. Derseh,

Thank you for submitting your manuscript to PLOS ONE. After careful consideration, we feel that it has merit but does not fully meet PLOS ONE’s publication criteria as it currently stands. Therefore, we invite you to submit a revised version of the manuscript that addresses the points raised during the review process. Please submit your revised manuscript by May 25 2024 11:59PM. If you will need more time than this to complete your revisions, please reply to this message or contact the journal office at plosone@plos.org. Please include the following items when submitting your revised manuscript:

We look forward to receiving your revised manuscript.

Kind regards,

Mesfin Gebrehiwot Damtew (PhD)

Academic Editor

PLOS ONE

Journal Requirements:

3. In the online submission form, you indicated that "All of the included data are available in the manuscript and further inquiries can be directed to the corresponding author when necessary."

4. Please include a copy of Table 1-5 which you refer to in your text on page 8-12.

5. We note you have included a table to which you do not refer in the text of your manuscript. Please ensure that you refer to Table 6-10 in your text; if accepted, production will need this reference to link the reader to the Table.

Reviewers' comments:

Reviewer's Responses to Questions

**Comments to the Author**

1. Is the manuscript technically sound, and do the data support the conclusions?

Reviewer #1: Yes

Reviewer #2: Partly

2. Has the statistical analysis been performed appropriately and rigorously? 

Reviewer #1: Yes

Reviewer #2: Yes

3. Have the authors made all data underlying the findings in their manuscript fully available?

Reviewer #1: Yes

Reviewer #2: Yes

4. Is the manuscript presented in an intelligible fashion and written in standard English?

Reviewer #1: Yes

Reviewer #2: No

5. Review Comments to the Author

Reviewer #1: The objective of this study is impressive and this may contribute a lot in understanding the open defecation in Ethiopia. However I have some major concerns that need to be addressed before we proceed with the publication possibilities. Below are my concerns-

1. The abstract and even the entire paper has not proposed solutions to effectively combat the practice of open defecation, solutions to remove the socio-cultural barriers associated with open defecation, solutions to bridge the spatial disparities in terms of open defecation in different regions of India. I recommend revising it accordingly.

2. The background section could be rewritten based on the above-revised examinations. Furthermore, the discussion could be reframed based on the revised objectives, policy strengths, and weaknesses. Additionally, information from census data could be included in the discussion section to better understand the open defecation scenario based on the whole population counts.

3. I would expect more explanations on why spatial clustering is important to be analyzed. As the author said, numerous studies have analyzed the associations between various variables (e.g. socioeconomic status, demographic characteristics of the individuals/households, etc.) and sanitation. Why do you think spatial clustering is worth of special attention?

4. Comparison with other countries such as India where research analyzed open defecation (Roy et al., 2023 in BMJ Open and Roy et al., 2023 in Global Transitions)

5. Recommendations could be more precise to help public interventions address the phenomenon, based on studied variables

6. Table 1-5 are missing, however, it cited in main text.

7. Age of the household head should be in ‘years’. please add in table

Statistical comments

1. Stating a moran’s I of 0.452 as a strong positive autocorrelation might not be appropriate. Kindly revisit.

2. Moran’s I close to 0.20 should not be considered to reflect the presence of spatial autocorrelation. Kindly revisit these points again and quote with proper reference if you still wish to proceed with the said findings

3. * are used possibly to explain the significance level however I did not find any clarity on this in the footnote of table. Need to be incorporated

4. p values written as 0.000000 should be reported as p<0.001. Kindly refer to the standard reporting guidelines by Bland and Altman BMJ series.

5. The present study presented OLS spatial regression because it was found to be the best-fit model based on the AIC values”??? So because u found OLS best fit using AIC, kindly present the AIC comparison of spatial as well as OLS model and then prove he model with best fit. Also, the Mornas I at the beginning itself reveals no spatial autocorrelation at multiple instances and that could be the reason that OLS is found to be the best fit. Kindly modify your study based on these findings and observation

6. R squared is not required if you report adjusted R squared. Jarque Bera test need to be interpreted to justify the usage of spatial analysis.

Reviewer #2: The authors examined a concerning public health topic, which could be valued for policy understanding. However, the analyses were very common. Therefore, I suggest authors to look into the issues more in-depth. I suggest authors read recent publications related to open defecation to revise their methods ((https://www.sciencedirect.com/science/article/pii/S2589791823000099, https://bmjopen.bmj.com/content/bmjopen/13/7/e072507.full.pdf)).

The World Health Organization (WHO) suggested that more than 1.5 billion people still do not have private toilets globally; out of these, 419 million still practice open defecation. I suggest authors add information on how the 1.5 billion OD practice varies region-wise like European region, African region, South Asian region, Southeast Asian region, and so on.

Please explain briefly how to construct the dependent variable, define it, and measure its limitations.

Explain each selected independent variable in tabular form. Cite some non-African studies where OD is a similar issue like in India (https://www.sciencedirect.com/science/article/pii/S2589791823000099, https://bmjopen.bmj.com/content/bmjopen/13/7/e072507.full.pdf).

The authors performed only Ordinary Least Squares (OLS) analysis. Why did they not perform a Spatial lag model and spatial error model? Please justify it.

Please add the limitations of the Geographically Weighted Regression (GWR) in the limitation section of the study.

The discussion section is poorly written. Make it more systematic and critical based on study findings. Try to highlight what is existing knowledge on OD and what are the new findings in your study. Try to compare to other non-African countries where OD prevalence is also high. Try to add more discussion on why spatial heterogeneity exists in your study, what are existing policies, and how spatial heterogeneity could be overcome by revising existing policies or introducing new policies.

I suggest authors to critique the ongoing policy like Community HEP based on present study findings and how it could revise the existing policy to achieve an open defecation-free country.

Minor comments

Please add the abbreviation of DALYs in the main text.

Please try to avoid the term "socio" twice in this sentence: "Demographic and socioeconomic characteristics of participants."

Please avoid using "we used"; replace it with "the study used".

Make all subheadings in the results section in concise form. All subheadings should look like sentences.

Please continue point one-digit reporting.

6. PLOS authors have the option to publish the peer review history of their article (what does this mean?). If published, this will include your full peer review and any attached files.

Reviewer #1: No

Reviewer #2: No

---

## [Author Response · Author response to Decision Letter 0]

22 May 2024

Authors’ point-by-point Responses to Editor’s comment for a manuscript ID: PONE-D-24-01346

Dear Editor, We would like to thank you for your scientific contribution by editing this manuscript. We have provided a point-by-point response for your comment:

https://journals.plos.org/plosone/s/file?id=wjVg/PLOSOne_formatting_sample_main_body.pdf, 

Authors responses: Dear Editor, thank you very much for your suggestion. We have confirmed that the manuscript format is based on the PLoS ONE style.

Authors’ responses: Dear Editor, thank you for this comment. We have improved the grammatical errors and language usage of the whole manuscript.

3. In the online submission form, you indicated that "All of the included data are available in the manuscript and further inquiries can be directed to the corresponding author when necessary."

Authors’ responses: Dear Editor, as you suggested, we have indicated this in the online submission.

4. Please include a copy of Table 1-5 which you refer to in your text on pages 8-12.

Authors responses: Dear Editor, thank you once again for your comment. Accordingly, we have included all of the tables immediately after their captions in the manuscript.

5. We note you have included a table to which you do not refer in the text of your manuscript. Please ensure that you refer to Tables 6-10 in your text; if accepted, production will need this reference to link the reader to the Table.

Authors’ responses: Dear Editor, We have revised the list of tables and figures as follow: there are 1–6 tables and 1–10 figures as cited in the manuscript

Authors’ point-by-point Responses to reviewer 1 comments -a manuscript ID: PONE-D-24-01346

Dear reviewer, We would like to thank you for reviewing this manuscript. We have provided a point-by-point response as follows: 

1. The abstract and even the entire paper have not proposed solutions to effectively combat the practice of open defecation, solutions to remove the socio-cultural barriers associated with open defecation, and solutions to bridge the spatial disparities in terms of open defecation in different regions of India. I recommend revising it accordingly.

 Authors’ responses: Dear reviewer, We would like to thank you for your scientific contribution by giving constructive suggestions to this manuscript. Based on your suggestion, we have modified the abstract as described on lines 41−46 and the discussion part by indicating different interventions and solutions on page 25, lines 552−557..

2. The background section could be rewritten based on the above-revised examinations. Furthermore, the discussion could be reframed based on the revised objectives, policy strengths, and weaknesses. Additionally, information from census data could be included in the discussion section to better understand the open defecation scenario based on the whole population counts

Authors responses: Dear reviewer, This is also an interesting comment, which helps further improve the quality of the manuscript. We have improved the introduction and the discussion better than in the previous submission.

3. I would expect more explanations on why spatial clustering should be analyzed. As the author said, numerous studies have analyzed the associations between various variables (e.g. socioeconomic status, demographic characteristics of the individuals/households, etc.) and sanitation. Why do you think spatial clustering is worth special attention?

Authors responses: Dear reviewer, We also appreciate your comment. We have modified and elaborated on the importance of geospatial analysis more, as it was described on pages 4 and 5, lines 103–112, in the revised manuscript.

4. Comparison with other countries such as India where research analyzed open defecation https://www.sciencedirect.com/science/article/pii/S2589791823000099
https://bmjopen.bmj.com/content/bmjopen/13/7/e072507.full.pdf

Authors responses: Dear reviewer, we would like to thank you for your kind suggestion to use the important references. We have used the indicated references, which improves the quality of this paper.

5. Recommendations could be more precise to help public interventions address the phenomenon, based on studied variables.

Authors’ responses: Dear reviewer, this is also an interesting comment. We have modified as this was displayed on page 02, lines 42–46 and on page 25, lines 552−557.

6. Tables 1-5 are missing, however, it cited in the main text.

Authors’ responses: Dear reviewer, Thank you for your comment. We have presented all tables in the proper place in the document.

7. The age of the household head should be in ‘years. please add in the table

Authors’ responses: Dear reviewer, Thank you for your nice comment. We have corrected it as it was displayed in Table 1 on page 7 and in Table 2 on page 10. 

Statistical comments

8. Stating a Moran’s I of 0.452 as a strong positive autocorrelation might not be appropriate. Kindly revisit.

Authors’ responses: Dear reviewer, Thank you for this comment. We have corrected it as displayed on page 12, line 252−254.

9. Moran’s, I close to 0.20 should not be considered to reflect the presence of spatial autocorrelation. Kindly revisit these points and quote with proper reference if you still wish to proceed with the said findings.

Authors’ responses: Dear reviewer, we appreciate your concern. Yet, the output of Global Moran’s I in this manuscript was I index = 0.45, Z-score = 9.88, p-value < 0.001. Since the Z-score is positive and the p-value is highly significant, we rejected the null hypothesis, which was random distribution, and we were compelled to accept the alternative hypothesis, which was spatial clustering.

10. * are used possibly to explain the significance level however I did not find any clarity on this in the footnote of the table. Need to be incorporated.

Authors’ responses: Dear reviewer, we are grateful for your comment. We used the (*) in Table 4 (OLS output) to indicate significant explanatory variables at a p-value less than or equal to 0.05. We have indicated this further in the footnote.

11. p values written as 0.000000 should be reported as p < 0.001. Kindly refer to the standard reporting guidelines by Bland and Altman BMJ series.

Authors’ responses: Dear reviewer, we would like to thank you for your comment. On page 12, table 4 and on page 17, on Table 5, we have amended it as you suggested.

12. The present study presented OLS spatial regression because it was found to be the best-fit model based on the AIC values”??? So because u found the OLS best fit using AIC, kindly present the AIC comparison of spatial as well as OLS models and then prove the model with the best fit. Also, the Moran’s I at the beginning itself reveals no spatial autocorrelation at multiple instances and that could be the reason that OLS is found to be the best fit. Kindly modify your study based on these findings and observation.

Authors’ responses: Dear reviewer, we would like to appreciate your concern. As we have described on page 16, lines 327–334, two of the six assumptions of the OLS model were not achieved. These were: based on the Jarque-Bera Statistic (a P-value < 0.05), residuals were not normally distributed, and because of this the OLS model prediction was biased. Additionally, according to the Koenker (BP) statistic, which is a significant result; this indicated non-stationarity. Therefore, we preferred GWR to OLS. For model comparison, we used AICc, and this was improved from the OLS model (-65.21) to GWR (-128.34). Similarly, the model performance of GWR (adjusted R = 72%) was a better fit than the OLS model (adjusted R = 63%). Therefore, GWR is the best-fit model. Even though, global Moran’s I statistics showed no clustering for residuals (I index = 0.07, Z-score = 1.51, p-value < 0.13), the other two assumptions were not met and the results are biased in OLS reg.

13. R squared is not required if you report adjusted R squared. Jarque Bera's test needs to be interpreted to justify the usage of spatial analysis.

Authors’ responses: Dear reviewer, thank you for your comments! Accordingly, we have omitted R squared on tables 4 and 5. Dear reviewer, on page 14, lines 327–329, we have put the interpretations of Jarque Bera's test.

Authors’ point-by-point Responses to reviewer 2 comments -a manuscript ID: PONE-D-24-01346

Dear reviewer, We would like to thank you for your scientific contribution by reviewing this manuscript. We have provided a point-by-point response as follows: 

1. I suggest authors read recent publications related to open defecation to revise their methods:

https://www.sciencedirect.com/science/article/pii/S2589791823000099
https://bmjopen.bmj.com/content/bmjopen/13/7/e072507.full.pdf

Authors’ responses: Dear reviewer, we would like to thank you for your constructive suggestions to improve the quality of this manuscript. Dear reviewer, we have used these articles as indicated in the introduction on page 3, lines 59−61, and in the discussion part on page 20, lines 417−418; on page 22, line 432; and on page 22, line 475, and on page 23 lines 497. 

2. The World Health Organization (WHO) suggested that more than 1.5 billion people still do not have private toilets globally; out of these, 419 million still practice open defecation. I suggest authors add information on how the 1.5 billion OD practice varies region-wise like European region, African region, South Asian region, Southeast Asian region, and so on.

Authors’ responses: Dear reviewer, We would like to thank you for your comments that helped us improve the quality of this manuscript. Based on your suggestion, we have added the global and regional information on pages 2 and 3, lines 54–68.

3. Please explain briefly how to construct the dependent variable, define it, and measure its limitations.

Authors’ responses: Dear reviewer, Thank you very much for this question. We have clarified as described on page 06, line numbers 159–161. It was a binary dichotomy, using “yes” for open defecation practice and “no” for those who have basic sanitation facilities.

4. Explain each selected independent variable in tabular form.

Authors’ responses: Dear reviewer, Thank you very much for your comment. We have presented all of the determinants as described on page 7 in Table 1.

5. Cite some non-African studies where OD is a similar issue like in India. https://www.sciencedirect.com/science/article/pii/S2589791823000099
https://bmjopen.bmj.com/content/bmjopen/13/7/e072507.full.pdf

Authors’ responses: Dear reviewer, We would like to thank you for this comment. We have added additional citations as indicated on pages 2 and 3, lines 53–61 in the introduction part, and on page 20, lines 417–418, on pages 22 line 475, and on page 23 line 497.

6. The authors performed only Ordinary Least Squares (OLS) analysis. Why did they not perform a spatial lag model and spatial error model? Please justify it.

Authors’ responses: Dear reviewer, Thank you very much for suggesting performing SLM and SEM. After fitting the GWR model, when we checked the clustering of residuals, it had no spatial dependence. Moreover, each of the coefficients was neither spatially autocorrelated nor had any spatial dependence. Because of these assumptions, we did not perform SLM and SEM.

7. Please add the limitations of the Geographically Weighted Regression (GWR) in the limitation section of the study. 

Authors’ responses: Dear reviewer, We thank you for this comment. We have added it on page 9, lines 226–228. 

8. The discussion section is poorly written. Make it more systematic and critical based on study findings. Try to highlight what is existing knowledge on OD and what are the new findings in your study. Try to compare to other non-African countries where OD prevalence is also high. Try to add more discussion on why spatial heterogeneity exists in your study, what are existing policies, and how spatial heterogeneity could be overcome by revising existing policies or introducing new policies.

Authors’ responses: Dear reviewer, We are grateful for this comment. Accordingly, we have improved the overall discussion part by comparing each of the findings with previous literature as well as providing possible justification.

9. I suggest authors to critique the ongoing policy like Community HEP based on present study findings and how it could revise the existing policy to achieve an open defecation-free country.

Authors’ responses: Dear reviewer, as suggested, we have incorporated HEP in the discussion. Health extension packages are the crucial tool to eliminate open defecation in Ethiopia, yet the implementation, monitoring, and evaluation systems are weak to achieve strategies. For example, we have included this on page 20, lines 431−432 and on page 21, line 448, and on page 22, lines 467−469.

Minor comments

10. Please add the abbreviation of DALYs in the main text.

Authors’ responses: Dear reviewer, we have added as it was described on page 03 and line 77.

11. Please try to avoid the term "socio" twice in this sentence: "Demographic and socioeconomic characteristics of participants."

Authors’ responses: Dear reviewer, thank you for your comment. We have corrected it accordingly.

12. Please avoid using "we used"; replace it with "the study used".

Authors’ responses: Dear reviewer, thank you very much for your interesting comment. We have modified it accordingly.

13. Make all subheadings in the results section in a concise form. All subheadings should look like sentences.

Authors’ responses: Dear reviewer, we have corrected all of the subheadings in the results section.

14. Please continue point one-digit reporting.

Authors’ responses: Dear reviewer, thank you for this comment. Most of the time, we use two-digit decimal numbers for presentation like frequencies, CI, OR, etc. We have reduced to two decimal points in to indicate level of significance.

---

## [Decision Letter · Decision Letter 1]

4 Jul 2024

Spatial variation and geographical weighted regression analysis to explore open defecation practice and its determinants among households in Ethiopia

PONE-D-24-01346R1

Dear Dr,

We’re pleased to inform you that your manuscript has been judged scientifically suitable for publication and will be formally accepted for publication once it meets all outstanding technical requirements.

Kind regards,

Mesfin Gebrehiwot Damtew (PhD)

Academic Editor

PLOS ONE

Additional Editor Comments (optional):

Reviewers' comments:

Reviewer's Responses to Questions

**Comments to the Author**

1. If the authors have adequately addressed your comments raised in a previous round of review and you feel that this manuscript is now acceptable for publication, you may indicate that here to bypass the “Comments to the Author” section, enter your conflict of interest statement in the “Confidential to Editor” section, and submit your "Accept" recommendation.

Reviewer #1: All comments have been addressed

2. Is the manuscript technically sound, and do the data support the conclusions?

Reviewer #1: Yes

3. Has the statistical analysis been performed appropriately and rigorously? 

Reviewer #1: Yes

4. Have the authors made all data underlying the findings in their manuscript fully available?

Reviewer #1: Yes

5. Is the manuscript presented in an intelligible fashion and written in standard English?

Reviewer #1: Yes

6. Review Comments to the Author

Reviewer #1: The authors have made all the necessary changes. The manuscript in its current form is suitable for publication.

7. PLOS authors have the option to publish the peer review history of their article (what does this mean?). If published, this will include your full peer review and any attached files.

Reviewer #1: **Yes: **AVIJIT ROY

---

## [Editor Report · Acceptance letter]

9 Jul 2024

PONE-D-24-01346R1 

PLOS ONE

Dear Dr. Derseh, 

I'm pleased to inform you that your manuscript has been deemed suitable for publication in PLOS ONE. Congratulations! Your manuscript is now being handed over to our production team.

Kind regards, 

on behalf of

Dr. Mesfin Gebrehiwot Damtew 

Academic Editor

PLOS ONE